# Incorporating Structured Representations into Pretrained Vision & Language Models Using Scene Graphs

**Roei Herzig**[* 1,3]**, Alon Mendelson**[*1]**,**
**Leonid Karlinsky**[4]**, Assaf Arbelle**[3]**, Rogerio Feris**[4]**, Trevor Darrell**[2]**, Amir Globerson**[1]

[1]Tel-Aviv University, [2]UC Berkeley, [3]IBM Research, [4]MIT-IBM Watson AI Lab

## Abstract

Vision and language models (VLMs) have demonstrated remarkable zero-shot (ZS) performance in a variety of tasks. However, recent works have shown that even the best VLMs struggle to capture aspects of compositional scene understanding, such as object attributes, relations, and action states. In contrast, obtaining structured annotations, such as scene graphs (SGs), that could improve these models is time-consuming and costly, and thus cannot be used on a large scale. Here we ask whether small SG datasets can provide sufficient information for enhancing structured understanding of pretrained VLMs. We show that it is indeed possible to improve VLMs when learning from SGs by integrating components that incorporate structured information into both visual and textual representations. For the visual side, we incorporate a special "SG Component" in the image transformer trained to predict SG information, while for the textual side, we utilize SGs to generate fine-grained captions that highlight different compositional aspects of the scene. Our method improves the performance of several popular VLMs on multiple VL datasets with only a mild degradation in ZS capabilities.

## 1 Introduction

In recent years, vision and language models (VLMs) such as CLIP (Radford et al., 2021) have demonstrated impressive results and extraordinary zero-shot capabilities when trained on massive datasets containing image-text pairs (e.g., LAION 400M Schuhmann et al., 2021). However, recent empirical studies (Thrush et al., 2022; Yuksekgonul et al., 2023; Ma et al., 2022) have shown that even the strongest VLMs struggle to perform compositional scene understanding, including identifying object attributes and inter-object relations.

Understanding the structure of visual scenes is a fundamental problem in machine perception and

has been explored extensively in many previous works (Xu et al., 2017; Herzig et al., 2020; Yang et al., 2022). In particular, datasets with scene graph (SG) annotations, such as the Visual Genome (VG) dataset (Krishna et al., 2017), have been collected and used to improve scene understanding models. However, such datasets are expensive to collect at scale and relatively small compared to those used in training VLMs.[1] This raises the following questions: (1) Can small datasets containing SG annotations be utilized to finetune VLMs and improve compositional scene understanding? (2) How should the model and training be adapted to best use this data? Here we show that it is indeed possible to improve VLMs using image-SG pairs by integrating components that incorporate structure into both visual and textual representations.

Our first step is to convert SGs into highly detailed captions. A naive approach would be to finetune VLMs on these image-text pairs, however, we have found that this approach does not sufficiently improve performance.[2] This is also aligned with recent work (Doveh et al., 2022; Yuksekgonul et al., 2023), showing that contrastive learning approaches allow the model to concentrate mainly on object labels disregarding other important aspects, such as relations and attributes. To alleviate this issue, we take inspiration from these works and use the SG to generate hard-negative captions that highlight structural aspects. For example, if an SG contains an edge "dog-chasing-cat", then we can reverse that edge into "cat-chasing-dog" and generate a corresponding negative caption.

Next, we turn to introduce structure into the visual representation. Inspired by prompt learning approaches (Jia et al., 2022; Herzig et al., 2022a; Zhou et al., 2022a), we incorporate into the image transformer encoder a set of "Adaptive Scene

---

[*]Equal contribution.

[1]VG contains $\sim 100K$ image-SG pairs, which is $\times 1000$ smaller than the large-scale VLMs pretraining datasets.

[2]See our ablations in section 4.5.

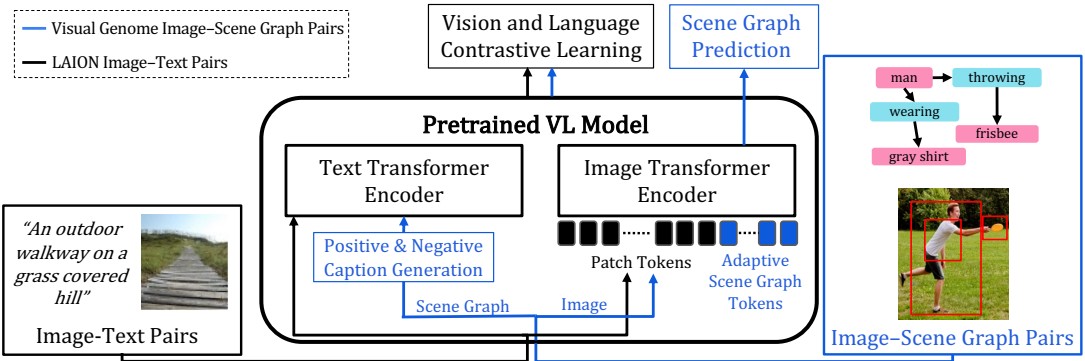

Figure 1: **Scene Graphs Improve Pretrained Vision-Language Models.** Vision and Language models (VLMs) are typically trained on large-scale image-text pairs (Left). We propose to improve pretrained VLMs by utilizing a small set of scene graph (SG) annotations (Right) from the Visual Genome dataset, that is richer and reflects structured visual and textual information. Specifically, we design a specialized model architecture and a new finetuning scheme when learning from SGs as follows: (i) Use the graph to generate fine-grained positive and negative captions that highlight different compositional aspects of the scene. (ii) Finetune the pretrained model using contrastive learning with the generated captions from the SGs, along with image-text pairs from LAION. (iii) Predict SG information (object, relations, and their coordinates) from an image by incorporating "Adaptive Scene Graph Tokens" into the image transformer encoder. During inference, these learned tokens, which capture structured information from the SG, interact with the patch tokens and CLS tokens to improve compositional scene understanding.

Graph Tokens", which interact with the patch and CLS tokens via attention. By training these tokens to predict SG information, the encoder can capture better structured representations.

However, the above task of predicting SG from the image transformer encoder is challenging, as it deviates from the initial VLM training objective. Toward this end, we designed a technique tailored to the SG tokens, which parameterized the SG tokens independently from the patch tokens. In particular, we modify the transformer layers throughout the network by decomposing the parameters into two distinct sets: SG and patch tokens. We have found that this allows better learning of the graph prediction task while still maintaining zero-shot performance. We name our proposed approach SGVL (*Scene Graphs for Vision-Language Models*). See Figure 1 for an overview.

To summarize, our main contributions are as follows: (i) We propose to exploit a small set of SG annotations that is rich with visual and textual information for enhancing compositional scene understanding in pretrained VLMs; (ii) We introduce a new finetuning scheme that captures structure-related information using the visual and textual components when learning from SG labels. Specifically, for the visual side, we incorporate special "Adaptive SG tokens" in the image transformer encoder, and train these to predict SG information; (iii) Our method shows improved performance on CLIP, BLIP, and BLIP2 on several benchmarks: Winoground (Thrush et al., 2022), VL-CheckList (Zhao et al., 2022), ARO (Yuksekgonul et al., 2023), and VSR (Liu et al., 2022),

highlighting the effectiveness of our approach.

## 2 Related Work

**Vision and Language Models**. In recent years, popular VLMs, such as CLIP (Radford et al., 2021), BLIP (Li et al., 2022b), BLIP2 (Li et al., 2023), and others, have shown impressive results and extraordinary zero-shot capabilities. These models are trained with image-text pairs to align the two modalities in a joint embedding space. However, recent empirical studies (e.g., VL-CheckList (Zhao et al., 2022), Winoground (Thrush et al., 2022), and ARO (Yuksekgonul et al., 2023)) have shown that these models do not perform well on tasks that require compositional understanding, including relations between objects, and their attributes. Specifically, it has been shown that VLMs tend to learn a "bag of objects" representation, leading them to be less structure-aware. Several recent works, such as NegCLIP (Yuksekgonul et al., 2023), Count-CLIP (Paiss et al., 2023), and SVLC (Doveh et al., 2022), haven shown that hard negative generation from image-text pairs using language augmentations could improve fine-grained understanding. Unlike these works, we utilize image-SG pairs to improve structured representations by predicting SG information from the image encoder.

**Multi-task Prompt Tuning**. The concept of prompt tuning for efficient finetuning of language models was introduced by Lester et al. (2021), and later explored in vision models (Jia et al., 2022; Wang et al., 2022) and VLMs (Ju et al., 2022; Zhou et al., 2022b). Several recent works (Asai et al., 2022; Sanh et al., 2022; Vu et al., 2022), have

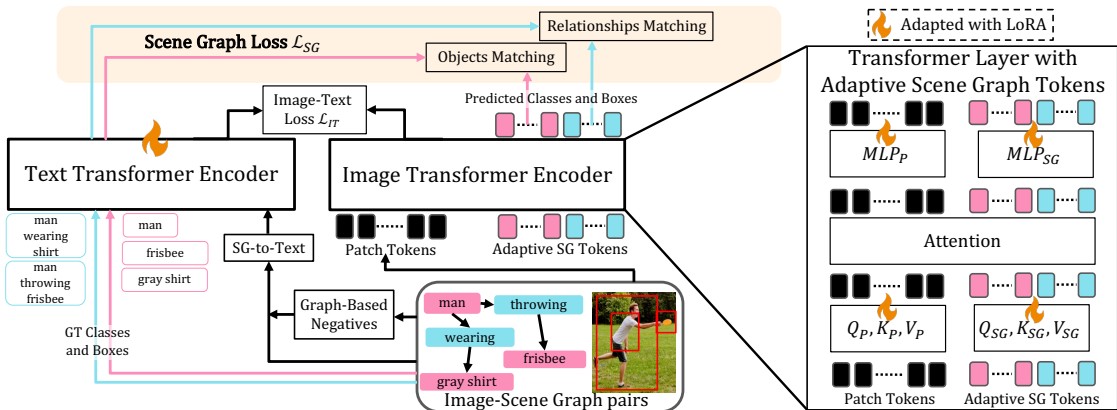

Figure 2: **Our Scene Graphs for Vision-Language Models (SGVL) Approach.** Our key goal is to capture structure-related information in both visual and textual components when learning from SGs. For the textual side, we generate captions and negative captions using the graph (Graph-Based Negatives and SG-to-text modules). For the visual side, we incorporate into the image transformer an additional set of learnable "Adaptive Scene Graph Tokens" to predict SG information. The transformer parameters are partitioned into two distinct sets, one for patch tokens and one for SG tokens (shown on the right). These tokens predict objects (pink tokens & arrows) and relationships (cyan tokens & arrows) from the image, and these predictions are matched with the ground-truth SGs via bipartite matching. To allow open vocabulary prediction, we embed the SG categories using the text transformer encoder. Last, we use LoRA adapters for finetuning, keeping all other model parameters frozen.

explored prompt tuning in the context of multi-task learning in NLP, followed by works in vision, and specifically in video domain (Avraham et al., 2022; Herzig et al., 2022a), and VL (Shen et al., 2022). Unlike these works, we add multiple prompts (which we refer to as Adaptive SG tokens) to VLMs in order to learn compositional scene information via the SG prediction task. We designed a technique tailored to the SG tokens, which decomposes the parameters throughout the transformer layers into two distinct sets: SG and patch tokens. This allows better learning of the graph prediction task while still maintaining zero-shot capabilities.

**Learning Structured Representations**. Structured representations have been shown to be beneficial in many applications: video understanding (Herzig et al., 2019, 2022b; Wang and Gupta, 2018), relational reasoning (Baradel et al., 2018; Battaglia et al., 2018; Jerbi et al., 2020), VL (Chen et al., 2020; Li et al., 2020; Tan and Bansal, 2019), human-object interactions (Kato et al., 2018; Xu et al., 2019), and even image & video generation (Bar et al., 2021; Herzig et al., 2020). In particular, SGs (Johnson et al., 2015; Xu et al., 2017; Herzig et al., 2018) have been extensively used to provide semantic representations in a wide range of applications (Johnson et al., 2018; Raboh et al., 2020; Yu et al., 2021; Cong et al., 2023). Unlike these previous works, here we propose a novel architecture design that utilizes scene graph data, demonstrating that a small amount of scene-graph data can be used to improve pretrained VLMs via fine-tuning. Finally, SG data has been recently

used to evaluate for compositional understanding in VLMs (Ma et al., 2022). This works differs from our approach, as this work only proposed a benchmark, and not a new training method, while our approach proposes a method for fine-tuning pretrained VLMs using scene graphs.

## 3 Scene Graphs for VL Models

We begin by describing the standard VL transformer architecture and the scene graph annotations (Section 3.1). We then introduce our structural components for both language (Section 3.2) and vision (Section 3.3), and the training losses (Section 3.4). Our method is illustrated in Figure 2.

### 3.1 Preliminaries

VLMs are typically trained with image-text pairs: $X_i = (I_i, T_i)$. Each of these modalities is processed by a separate encoder, and the training objective is to map the embeddings using contrastive learning. Next, we briefly describe the encoders.

**Language Transformer Encoder** $E_T$. The text encoder is a transformer (Vaswani et al., 2017) as described in CLIP and others, where a CLS token is appended to the beginning of the text, and the final CLS embedding is used as the text embedding.

**Vision Transformer Encoder** $E_I$. A typical vision transformer model (Dosovitskiy et al., 2021) takes an image $I$ as input, extracts $N$ non-overlapping patches,[3] and projects them into a lower-dimension $d$, followed by adding spatial position embeddings,

---

[3]We refer to these patches as "patch tokens".

resulting in a new embedding $z_i$. This forms the input tokens to the vision transformer encoder:

$$z = [z_{CLS}, z_1, z_2, \cdots, z_N] \qquad (1)$$

where $z_{CLS}$ is a learnable token. The input $z$ is fed into a standard transformer, and the final representation of the CLS token is the image embedding.

**Scene Graphs (SGs)**. Our motivation is to improve VLMs through structured annotations from an SG dataset. Formally, an SG is a tuple $G = (V, E)$ defined as follows: *(i) Nodes $V$* - A set of $n$ objects. Every object in the SG contains a class label, a bounding box, and attributes. *(ii) Edges $E$* - A set of $m$ edges. These relationships are triplets $(i, e, j)$ where $i$ and $j$ are object nodes, and $e$ is the category of the relation between objects $i$ and $j$. More details of the SG preprocessing are in Section B.1.

**Problem Setup**. As mentioned above, we use image-SG pairs $(I_G, G)$ from VG during finetuning to improve structure understanding along with standard LAION image-text pairs $(I, T)$. Next, we describe the textual and visual components that capture structure in both modalities.

## 3.2 Structural Language Component

We begin by describing how we transform an SG into text and then explain how to manipulate the SG with our Graph-Based Negatives to further capture the structure in the model. For a visualization, see Figure 7 in the supplementary.

**Scene Graph-to-Text**. Given an image $I$ and a corresponding SG, $G$, we use $G$ to generate a textual caption for the image. We iterate over the connected components of $G$ one by one. For each component, we iterate over the edges, and for each edge between object nodes $o_1$ and $o_2$ with relation $r$, we generate the text $o_1, r, o_2$ (e.g., "cat chasing dog"). If a node has an attribute, we prepend it to the node's object category. Last, we generate a single caption by concatenating the captions of the connected components separated by a period.

**Graph-Based Negatives (GN)**. We have found that using the scene graph data solely as image-text pairs with contrastive loss is not enough to force the model to develop structural understanding. As shown in recent work (Yuksekgonul et al., 2023), the commonly used contrastive learning allows the model to concentrate mainly on object labels disregarding other important aspects, such as relations and attributes. In order to provide more focus on such aspects, we exploit the SG structure

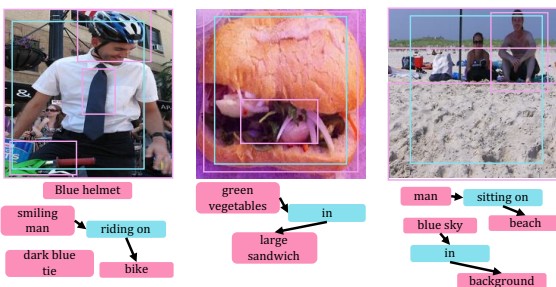

Figure 3: **"Adaptive SG Tokens" Visualization**. The predictions of object tokens (pink) and relationship tokens (cyan) are shown for images that are not in the VG training data.

and propose a set of predefined graph-based rules (See Section B.2) that modify SGs and make them semantically inconsistent with the image. Next, these SGs are transformed into negative textual captions, which are used with a specified loss to motivate the model to focus on structural aspects.

## 3.3 Structural Visual Component

**Scene Graphs Tokens**. We propose to capture structure-related information in the image encoder by predicting SGs. Toward this end, we add a set of "SG tokens" that are learned prompts designed to predict objects and relationships. The SG tokens consist of two groups: (i) "object tokens" that represent objects, their locations and their attributes, and (ii) "relationship tokens" that represent relationships and their locations.

Formally, we define a fixed set of $\tilde{n}$ learned object prompts and denote them by $p_1^o, p_2^o, \cdots, p_{\tilde{n}}^o \in \mathbb{R}^{1 \times d}$. Similarly, we define a fixed set of $\tilde{m}$ relationships prompts and denote them by $p_1^r, p_2^r, \cdots, p_{\tilde{m}}^r \in \mathbb{R}^{1 \times d}$. We refer to these prompts as the learned SG tokens. The SG tokens are concatenated with the standard CLS and patch tokens to obtain the following inputs to the transformer:

$$z = [z_{CLS}, z_1, ..., z_N, p_1^o, ..., p_{\tilde{n}}^o, p_1^r, ..., p_{\tilde{m}}^r] \quad (2)$$

Next, the transformer processes the input $z$, resulting in a new representation for each token. We use the new SG tokens representations to predict object and relationship labels and localization, by passing them through two feed-forward networks (FFNs). These predictions are supervised with ground-truth SG annotations through a matching process. Figure 3 visualizes the SG tokens learned by our model. More details are in Section 3.4.

**Adaptive SG tokens**. We have found that the SG prediction task is challenging as it deviates from the initial VLM training objective. To alleviate

this issue, we consider a modification of the image transformer tailored specifically to the SG tokens. Specifically, we decompose the parameters throughout the transformer layers into two distinct sets: SG and patch tokens. This allows better learning of the graph prediction task. In more detail, recall that transformer layers have matrices for mapping tokens to queries, keys, and values. We denote these existing matrices for the patch tokens by $Q_P, K_P, V_P$. We introduce a separate set of matrices $Q_{SG}, K_{SG}, V_{SG}$ that is used with the SG tokens. Importantly, the attention is performed over *all* tokens (patch and SG). Similarly, for the MLP component in the transformer layer, we also have a different version for the patch tokens ($\text{MLP}_P$) and the SG tokens ($\text{MLP}_{SG}$).

**Parameter Efficiency**. To perform efficient finetuning, we use LoRA adapters for the VLM. Specifically, for each trainable matrix $W \in \mathbb{R}^{u \times v}$ (e.g., $W = Q_P$ or $W = Q_{SG}$), we let $W_0$ denote its pretrained weights, and parameterize the learned matrix as:

$$W = W_0 + AB \quad (3)$$

where $A \in u \times r$ and $B \in r \times v$ are $r$-rank matrices and $r$ is a hyperparameter.[4] We note we use two distinct $r$: $r_p$ and $r_{SG}$ for weights associated with the patch and SG tokens (as described above), respectively. During training, $W_0$ is kept frozen, while $A, B$ are learned. Overall, our additional trainable parameters are only 7.5% of the model.

**Open Vocabulary SG Prediction**. The SG tokens predict the annotated SGs category information for objects, relationships, and attributes. A naive implementation of this idea would require a prediction head for each category. However, the VG dataset contains approximately 70K object categories and 40K relationship categories, and thus poses a significant challenge. Previous work (Xu et al., 2017) introduced a split containing only 100 object labels and 50 relation labels, which limits the dataset. Rather than restricting our data in this way, we use an open vocabulary approach. Namely, we utilize the text encoder to embed the categories from the SG components. Next, we use these embeddings in training to supervise the SG tokens. For example, if node $i$ has category "dog" and attribute "black", we train one of the object tokens to predict the embedding of the phrase "black dog". This also applies to the prediction of relationship tokens, e.g.,

a relationship token predicts the embedding of the phrase "dog chasing cat". More details are in the next section.

### 3.4 Training and Losses

During training our batch contains image-text pairs $(I, T)$ along with image-SG pairs $(I_G, G)$. We use the latter to generate positive captions $(T_P)$ and negative captions $(T_N)$. We finetune our model using these inputs while optimizing the losses below.

**Image-Text Loss**. Our image-text loss is comprised of *contrastive loss* and *graph negatives loss*. *Contrastive Loss:* We apply contrastive loss on image-text pairs, as in Radford et al. (2021). Hence, the loss is calculated as follows based on standard pairs and those generated from SGs:

$$\mathcal{L}_{Cont} := \text{Contrastive}(E_I(\tilde{I}), E_T(\tilde{T})) \quad (4)$$

where $\tilde{I} = I \cup I_G$ and $\tilde{T} = T \cup T_P$. *Graph-Based Negative Loss:* For each image-SG pair we apply a loss that drives the embedding of $I_G$ to be more similar to that of $T_P$ than $T_N$:

$$\mathcal{L}_{GN} := \sum_{I_G, T_P, T_N} -log \left( \frac{e^{C(I_G, T_P)}}{e^{C(I_G, T_P)} + e^{C(I_G, T_N)}} \right)$$

where $C(\cdot, \cdot)$ is the cosine similarity between the image and text embeddings. Finally, the image-text loss is a weighted sum of both losses:

$$\mathcal{L}_{IT} := \mathcal{L}_{Cont} + \lambda_{GN} \mathcal{L}_{GN} \quad (5)$$

and $\lambda_{GN}$ is a hyperparameter. For more information, see Section B.4 in supplementary.

**Scene Graph Loss**. The graph $G$ contains several annotations: the set of object categories $O$, their set of bounding boxes, the set of relationship categories $R$, and their bounding boxes. As we do not aim to predict the object and relationship categories directly, but rather use the embeddings from the VLM, we extract the category embeddings with the text encoder $E_T$: $\tilde{O} = E_T(O) \in \mathbb{R}^{(n+1) \times d}$, and $\tilde{R} = E_T(R) \in \mathbb{R}^{(m+1) \times d}$. We note that $(n+1)$ and $(m+1)$ classes are due to "no object" and "no relationship" classes. These class embeddings together with the bounding boxes are the SG elements that we aim to predict from the SG tokens.

We next describe the prediction process. The image encoder outputs a set of object tokens and a set of relationship tokens. We apply two separate FFNs to predict bounding boxes and class embeddings. To calculate probabilities over $n + 1$ and

---

[4]For the SG-token parameters, we set $W_0$ to the value of the corresponding parameters of the patch tokens.

| Model | Winoground | | | VL-Checklist | | | ARO | | VSR | ZS |
|---|---|---|---|---|---|---|---|---|---|---|
| | All Dataset | | | All Datasets Avg | | | Flickr30K Order | COCO Order | All Dataset Avg | 21 Tasks Avg |
| | Text | Image | group | Attribute | Object | Relation | | | | |
| FLAVA | 32.3 | 20.0 | 14.5 | 54.6 | 70.6 | 46.6 | 12.9 | 3.9 | 54.1 | - |
| ViLT | 34.8 | 14.0 | 9.3 | 73.3 | 85.0 | 62.0 | 22.4 | 18.7 | - | - |
| LLaVA | 24.8 | 25.0 | 13.0 | 65.5 | 83.1 | 83.0 | 98.1 | 97.5 | - | - |
| MiniGPT-4 | 23.3 | 18.0 | 9.5 | 71.3 | 84.2 | 84.1 | 99.4 | 98.9 | - | - |
| CLIP | 30.7 | 10.5 | 8.0 | 65.5 | 80.6 | 78.0 | 59.5 | 46.0 | - | 56.4 |
| BLIP | 39.0 | 19.2 | 15.0 | 75.2 | 82.2 | 81.5 | 27.9 | 24.9 | 56.5 | 49.0 |
| BLIP2 | 42.0 | 23.8 | 19.0 | 77.8 | 84.9 | 84.9 | 33.9 | 32.3 | 61.9 | 52.5 |
| NegCLIP | 29.5 | 10.5 | 8.0 | 68.0 | 81.4 | 81.3 | 91.0 | 86.0 | - | 55.1 |
| NegBLIP | 42.5 | 24.0 | 18.5 | 78.2 | 83.7 | 81.9 | 85.0 | 84.7 | 57.9 | 48.2 |
| NegBLIP2 | 41.5 | 26.0 | 20.5 | 79.0 | 87.0 | 88.2 | 91.8 | 88.2 | 62.1 | 51.7 |
| CLIP-SGVL (ours) | 32.0 (+1.3) | 14.0 (+3.5) | 9.8 (+1.8) | 72.0 (+6.5) | 82.6 (+2.0) | 82.0 (+4.0) | 82.0 (+22.5) | 78.2 (+32.2) | - | 54.3 (-2.1) |
| BLIP-SGVL (ours) | 42.8 (+3.8) | 27.3 (+8.1) | 21.5 (+6.5) | 81.8 (+6.6) | 85.2 (+3.0) | 81.9 (+0.4) | 70.0 (+42.1) | 71.0 (+46.1) | 62.4 (+5.9) | 48.0 (-1.0) |
| BLIP2-SGVL (ours) | 42.8 (+0.8) | 28.5 (+4.5) | 23.3 (+4.3) | 81.2 (+3.4) | 88.4 (+3.5) | 88.8 (+3.9) | 77.0 (+43.1) | 77.0 (+44.7) | 63.4 (+1.5) | 51.4 (-1.1) |

Table 1: **Winoground, VL-Checklist, ARO, VSR, and Zero-Shot (ZS) Results**. Gains & losses are relative to the base models.

$m + 1$ classes, we compute the cosine similarity of the predicted class embeddings with the GT class embeddings, $\tilde{O}$ and $\tilde{R}$ followed by a softmax.

Next, to determine which SG tokens correspond to which GT objects and relationships, we match the predictions of the SG tokens with the ground-truth SG. We follow the matching process and loss computation as in DETR (Carion et al., 2020), except that in our case, objects and relationships are matched separately. Our final SG loss $\mathcal{L}_{SG}$ is the sum of the objects matching loss $\mathcal{L}_O$ and the relationships matching loss $\mathcal{L}_R$. The complete matching process and losses details are in Section B.3.

We optimize our final loss as the sum of the scene graph loss and the image-text loss:

$$\mathcal{L}_{Total} := \mathcal{L}_{IT} + \lambda_{SG}\mathcal{L}_{SG} \qquad (6)$$

where $\lambda_{SG}$ is a hyperparameter.
re

## 4 Experiments and Results

We apply our SGVL approach to three popular VLMs: CLIP, BLIP, and BLIP2.[5] We evaluate on several VL compositional datasets following the standard protocol for each model. Additional results and ablations are in Supp. Section A.

### 4.1 Datasets

We describe the training and evaluation datasets below. More details are in Supp. Section D.
**Training**. For training, we use image-SG data from Visual Genome (VG), along with a small subset (less than 1%) of the LAION dataset as "standard" image-text pairs. **VG** is annotated with $108,077$

---
[5]More details of BLIP2 modifications are in Section B.5.

images and corresponding SGs, and **LAION 400M** is a large-scale image-text pair dataset that was automatically curated from the Internet.

**Evaluation**. We evaluate our method on four main benchmarks for compositional scene understanding: VL-Checklist (VLC) (Zhao et al., 2022), Winoground (Thrush et al., 2022), ARO (Yuksekgonul et al., 2023), VSR (Liu et al., 2022), and zero-shot classification. **(1) VLC** combines samples from the following datasets: VG (Krishna et al., 2017), SWiG (Pratt et al., 2020), VAW (Pham et al., 2021), and HAKE (Li et al., 2019). For each image, two captions are given, a positive and a negative. The negative caption is constructed by modifying one word in the positive caption that corresponds to a structural visual aspect (e.g., attribute). We report results on a combined VLC dataset excluding VG. **(2) Winoground** probes compositionality in VLMs. Each sample is composed of two image-text pairs that have overlapping lexical content but are differentiated by swapping an object, a relation, or both. For each sample, two text-retrieval tasks (text score), and two image-retrieval tasks (image score) are defined. The group score represents the combined performance. A recent study (Diwan et al., 2022) has shown that solving Winoground requires not just compositionality but also other abilities. The study suggested a subset (NoTag) that solely probes compositionality. We report results on the full dataset and the NoTag split. **(3) ARO** proposes four tasks that test sensitivity to order and compositionality: VG Relation, VG Attribution, COCO & Flickr30k Order. Since our approach is trained on VG, we report only the COCO and Flickr30k order tasks. **(4) VSR** tests spatial understanding. The

| | Winoground | | | VL-Checklist | | | | | | | |
|---|---|---|---|---|---|---|---|---|---|---|
| | | NoTag | | | Attribute | | | | Object | | Relation |
| Model | Text | Image | Group | Action | Color | Material | Size | Location | Size | Action |
| CLIP | 30.4 | 11.1 | 8.2 | 68.1 | 70.2 | 73.1 | 52.9 | 81.0 | 80.1 | 78.0 |
| BLIP | 44.8 | 23.8 | 19.2 | 79.5 | 83.2 | 84.7 | 59.8 | 83.0 | 81.3 | 81.5 |
| BLIP2 | 50.0 | 31.9 | 26.7 | 81.0 | 86.2 | 90.3 | 61.7 | 85.4 | 84.3 | 84.9 |
| CLIP-SGVL(ours) | 33.3 (+2.8) | 14.0 (+2.9) | 8.7 (+0.5) | 76.6 (+8.5) | 78.7 (+8.5) | 81.3 (+8.2) | 59.7 (+6.8) | 83.2 (+2.2) | 82.0 (+1.9) | 81.3 (+3.3) |
| BLIP-SGVL(ours) | 45.9 (+1.1) | 34.3 (+10.5) | 25.6 (+6.4) | 79.2 (-0.3) | 94.5 (+11.3) | 91.9 (+7.2) | 73.3 (+13.5) | 86.4 (+3.4) | 83.9 (+2.6) | 81.9 (+0.4) |
| BLIP2-SGVL(ours) | 51.7 (+1.7) | 37.2 (+5.3) | 29.0 (+2.3) | 82.4 (+1.4) | 91.7 (+5.5) | 92.2 (+1.9) | 70.1 (+8.4) | 89.0 (+4.6) | 87.6 (+3.3) | 88.8 (+3.9) |

Table 2: **Winoground and VL-Checklist Results**. Results for SGVL and baselines on VL-Checklist subsets and the NoTag split of Winoground. For VL-Checklist we exclude the Visual Genome dataset.

| Model | Adjacency | Directional | Orientation | Projective | Proximity | Topological | Unallocated | Average |
|---|---|---|---|---|---|---|---|---|
| BLIP | 55.4 | 48.9 | 53.7 | 58.2 | 56.5 | 55.6 | 63.4 | 56.5 |
| BLIP2 | 56.2 | 47.9 | 59.8 | 62.5 | 55.8 | 66.7 | 66.3 | 61.9 |
| BLIP-SGVL (ours) | 57.7 (+2.3) | 54.1 (+5.2) | 57.8 (+4.1) | 63.8 (+5.6) | 57.8 (+1.3) | 64.8 (+8.8) | 68.8 (+5.4) | 62.4 (+5.9) |
| BLIP2-SGVL (ours) | 59.8 (+3.6) | 56.9 (+9.0) | 58.5 (-1.3) | 61.5 (-1.0) | 59.7 (+3.9) | 70.0 (+3.3) | 66.8 (+0.5) | 63.4 (+1.5) |

Table 3: **VSR Results.** Results for SGVL and baselines on the VSR dataset.

dataset consists of pairs of an image and a description of the spatial relations between two objects shown in the image. The VLM should classify the pairs as either true or false. We do not evaluate CLIP-based models here as CLIP does not allow such classification in a straightforward manner.

**Zero-Shot (ZS) Classification**. We evaluate 21 classification datasets following the protocol from ELEVATER (Li et al., 2022a). The evaluation includes datasets such as ImageNet, CIFAR100, and others. We report the average results over the 21 tasks in Table 1 and Table 4b in Ablations.

## 4.2 Implementation Details

We implemented SGVL using Pytorch. For code and pretrained models, visit the project page at `https://alonmendelson.github.io/SGVL/`. Our code and training procedures are based on CLIP, BLIP and BLIP2. For CLIP, we use the ML-Foundation Open-CLIP repository (Ilharco et al., 2021), and for BLIP/BLIP2 we use the official implementations.[6] We use the ViT/B-32 model architecture for CLIP, ViT/B-16 for BLIP and ViT-g for BLIP2. The models are initialized with original weights released by the respective authors. The training batch contains 256/32/32 image-text pairs and 8 image-SG pairs for CLIP/BLIP/BLIP2-SGVL respectively. For more info, please refer to Supp. Section D. Last, we select the best checkpoint for each experiment using the validation set, which is composed of VG samples from VLC that are not in the test set (as mentioned above, we report VLC results excluding VG).

---
[6] `https://github.com/salesforce/LAVIS`

## 4.3 Baselines

In the experiments, we compare our SGVL approach to three VLMs: CLIP (Radford et al., 2021), BLIP (Li et al., 2022b), and BLIP2 (Li et al., 2023) to which SGVL was applied. Additionally, we compare with NegCLIP (Yuksekgonul et al., 2023), which uses negative text augmentations and also demonstrated strong results on these datasets. For a fair comparison, we also implement BLIP/BLIP2 versions of the NegCLIP approach, which we refer to as NegBLIP/NegBLIP2. Last, we also show comparison with the state-of-the-art VLMs: FLAVA (Singh et al., 2021) and ViLT (Kim et al., 2021), as well as newly introduced generative VLMs (GVLMs), such as LLaVA (Liu et al., 2023) and MiniGPT-4 (Zhu et al., 2023).

## 4.4 Results

Results are shown in Table 1, and demonstrate that CLIP/BLIP/BLIP2-SGVL outperforms the pretrained base models across several datasets. These improvements come at the price of a slight degradation in zero-shot performance. This may be due to several factors: (1) Finetuning with negative captions may harm the ZS performance, as observed in Yuksekgonul et al. (2023); (2) Finetuning for SG prediction deviates from the original VLM training objective; (3) We use only 1% of LAION image-text pairs for finetuning. Additionally, our method outperforms the Neg baselines on all datasets except ARO, which measures the sensitivity to text order. For this task, the text augmentations in Yuksekgonul et al. (2023) are more appropriate. Finally, we note that we do not evaluate CLIP-based models on VSR (See VSR in Section 4.1).

| (a) Scene Graph Utilization | | | | (b) Adaptive SG Tokens | | | | (c) Sparse Vs. Dense SGs | | | |
|---|---|---|---|---|---|---|---|---|---|---|---|
| Model | Text | Image | Group | Model | WG | ZS | mAP | Model | Text | Image | Group |
| BLIP | 39.0 | 19.2 | 15.0 | BLIP | 15.0 | 49.0 | - | 30% of Graph | 33.2 | 26.5 | 18.0 |
| BLIP+Graph Text (GT) | 40.3 | 20.5 | 16.5 | BLIP + SG Tokens | 20.0 | 47.5 | 16.1 | 70% of Graph | 40.7 | 26.5 | 20.0 |
| BLIP+GT+Graph Neg. (GN) | 40.5 | 25.5 | 19.0 | BLIP + Adap. SG Tokens | 21.5 | 48.0 | 17.7 | w/o Relations | 40.5 | 20.5 | 15.8 |
| BLIP+GT+GN+SG Tokens | 42.8 | 27.3 | 21.5 | | | | | Entire Graph | 42.8 | 27.3 | 21.5 |

Table 4: **Ablations on the Winoground Dataset.** We show (a) The contribution of our proposed components, utilizing the SG information. (b) The benefits of our adaptive SG tokens. Reported metrics are: Winoground group score (WG), zero-shot (ZS) on ELEVATER, and mAP of SG prediction. (c) Importance of the image-SG comprehensiveness. More ablations are in Section A.

Table 2 and Table 3 show the performance of our method on fine-grained Winoground, VLC, VSR splits. For most splits, our method is significantly better or comparable to the pretrained models. In Figure 4, we show where our model improves upon the baseline and where it still fails. For more results, please refer to Section A.3.

### 4.5 Ablations

We perform a comprehensive ablation on the Winground dataset with our SGVL approach using the BLIP model[7] (see Table 4). More ablations are in Section A.1. We also provide ablations on *all datasets* in Section A.2.

**Finetuning on VG without SGs**. We compare our approach to naive finetuning with captions provided by the VG dataset. We finetuned BLIP on textual descriptions from VG, resulting in 40.0/20.5/16.0 for Winogroundf Text/Image/Group scores, while our BLIP-SGVL achieves 42.8/27.3/21.5, indicating the improvements are not solely due to the VG data.

**SG prediction without SG tokens**. Our model predicts SG information from specialized SG tokens. A naive approach might have been to predict from the CLS token. Thus, we consider a variant that we refer to as *BLIP MT (multi-task)*, which is a simple implementation of object and relation prediction. This variant does not include SG tokens and instead predicts the graph using object and relation MLPs on top of the CLS token. Using exactly the same training recipe and data as our BLIP-SGVL, this ablation achieves 38.5/22.5/17.5 for Winoground Text/Image/Group scores, while our BLIP-SGVL achieves 42.8/27.3/21.5, justifying our design choice of SG tokens.

**Scene graph utilization ablation**. Our model consists of three parts: converting SG into captions, adding hard-negatives, and adding adaptive SG tokens. In Table 4a, we examine the con-

tribution of each component. By finetuning the model with only positive captions generated from the SGs (*BLIP+GT*), we obtain +1.3/+1.3/+1.5 for Winoground Text/Image/Group scores over the BLIP baseline. When the graph-based negatives (*BLIP+GT+GN*) are added, an improvement of +1.5/+6.3/+3.2 is achieved compared to the baseline. Finally, when fully implementing our SGVL approach, i.e. adding the SG prediction task from the SG tokens, we achieve improved results over the baseline of +3.8/+8.1/6.5. This highlights the contribution of the model components.

**Adaptive scene graph tokens**. Our approach uses different transformer parameters for SG and patch tokens, unlike the standard transformer, which uses the same parameters for all tokens. Thus, we next examine what happens when SG and patch tokens use the same parameters. In Table 4b, we report the performance of two variants on three tasks: Winoground group score (WG), Zero-shot classification (ZS) on ELEVATER, and SG prediction (mAP metric). The first variant, *BLIP + SG Tokens*, refers to the addition of tokens dedicated to predicting SGs with the same transformer parameters shared between them and other input tokens. The second variant, *BLIP + Adaptive SG Tokens*, refers to our technique that introduces parameters specific to the SG tokens in every transformer layer (see section 3.3). We can see that the second variant outperforms the SG token addition in all tasks. This demonstrates how our modification to the image transformer encoder improves SG prediction and VL performance without compromising ZS.

**Using sparse vs. dense SGs**. In this ablation, we investigate whether the density of the SG in VG affects performance, since SGs contain denser and richer information than standard captions. Towards this end, we train SGVL with sparsified versions of the graph. Specifically, we train two variants where objects and relations from the graphs are randomly removed (30% and 70%) and a third variant in which all relations are removed. As can be seen

---

[7]We chose BLIP due to its lower computational requirements (e.g., BLIP2 requires A100).

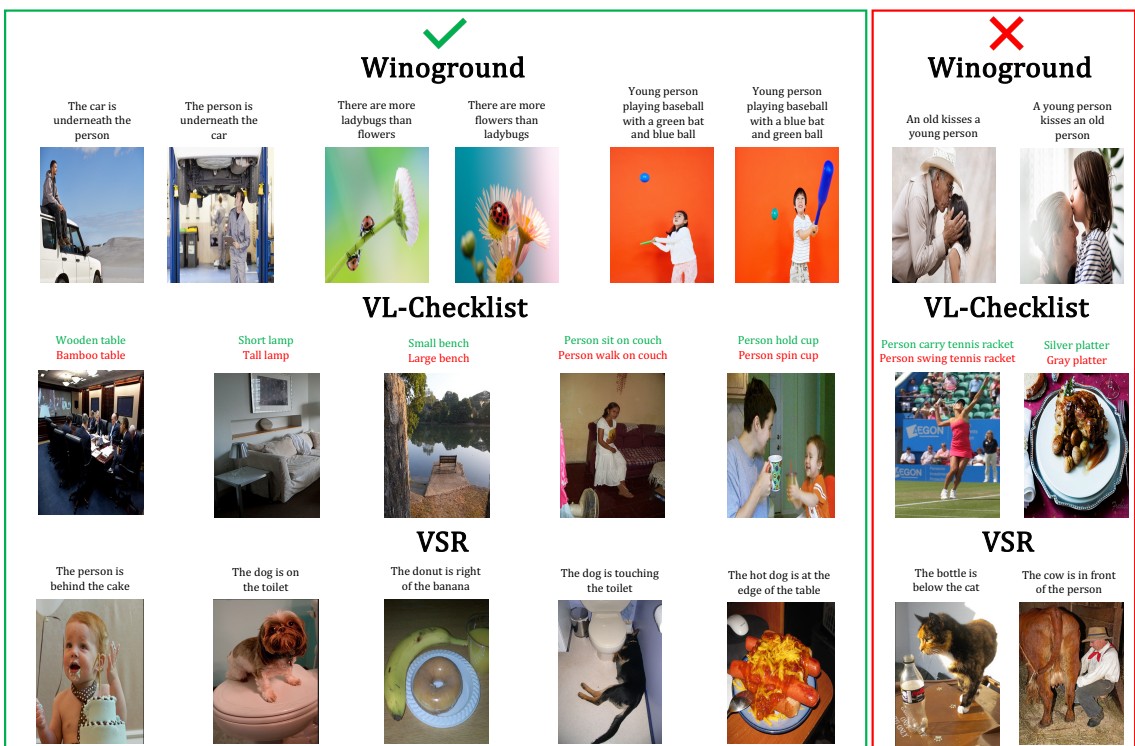

Figure 4: **Predictions visualization on Winoground, VL-Checklist, and VSR**. The left panel shows where our model succeeds and the baseline fails. The right panel shows where our model still fails. For VL-Checklist examples, true captions are in **green** and false captions in **red**. Our model outperforms the baseline in samples that require understanding relationships between objects and binding attributes to objects. The failure cases illustrate the complexity and ambiguity of the samples.

in Table 4c, our results show that our model performs better when the graphs are denser, richer, and describe the image more accurately, highlighting the motivation to utilize SGs for VLMs.

**Training without LAION image-text pairs**. In SGVL, we train simultaneously with image-text pairs from LAION and image-SG pairs from VG. To test the effectiveness of simultaneous training, we train only with Image-SG pairs and obtain a degradation of -2.0/-1.2/-1.2 for Text/Image/Group scores compared to our BLIP-SGVL. In addition, we observe a degradation of 1.1% in ZS performance compared to our BLIP-SGVL model. This indicates that simultaneous training is beneficial.

## 5 Conclusions

Structured understanding of complex scenes is a key element of human perception, but its modeling still remains a challenge. In this work, we propose a new approach for incorporating structured information into pretrained VLMs from SG data to improve scene understanding. We demonstrate improved performance on four benchmarks probing compositional scene understanding with only a mild degradation in ZS performance. Our findings suggest that only a small amount of high-quality annotations may be sufficient to improve such mod-

els. We hope these findings will encourage future work to generalize our approach beyond the VL regime or using other types of dense annotations, such as segmentation maps and depth maps.

## 6 Limitations

As mentioned above, our work proposes a specialized model architecture and a new finetuning scheme for learning from SGs. We demonstrate improved performance on several different models and datasets. Nevertheless, our work has some limitations. First, we leverage SG annotations since they are rich and reflect structured visual and textual information. However, the quality of the data is crucially important to our method, and thus poor data may result in lower performance. Finally, our improvements may be restricted by the fact that these annotations are rare and expensive to collect.

### Acknowledgements

This project has received funding from the European Research Council (ERC) under the European Unions Horizon 2020 research and innovation programme (grant ERC HOLI 819080). Prof. Darrell's group was supported in part by DoD, including PTG and/or LwLL programs, as well as BAIR's industrial alliance programs.

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

## Supplementary Material for "SGVL"

Here we provide additional information about our experimental results, qualitative examples, implementation details, and datasets. Specifically, Section A provides more experiment results, Section B provides more additional method details, Section C provides qualitative visualizations to illustrate our approach, and Section D provides additional implementation details.

## A    Additional Experiment Results

We begin by presenting several additional ablations (Section A.1) that further demonstrate the benefits of our SGVL approach. Next, we present the BLIP model ablations for all datasets (Section A.2). Last, we present additional results (Section A.3).

### A.1    Additional Ablations

In what follows, we provide additional ablations that further illustrate the benefits of SGVL. For all ablations, we finetuned the BLIP model using the same recipe, in which only LoRA adapters are learned, and the training batch consists of VG image-SG pairs and LAION image-text pairs.

**The importance of the SG data**. To examine the significance of the information provided by scene graphs, we suggest learning SGs without any useful information. Thus, we run an experiment in which the SGs are completely random. This ablation obtains on Winoground 37.8/18.7/14.0 compared to our BLIP-SGVL 42.8/27.3/21.5 for Text/Image/Group scores (the BLIP baseline obtains 39.0/19.2/15.0). This illustrates that the approach we employ is not merely a regularization, but also provides important information about the SGs that can be used by pretrained VLMs.

**The effect of image-SG data size**. In this experiment, we train our method using varying amounts of image-SG pairs of data (10%, 40%, 70%, and 100% of the dataset) in order to examine the effect of the data portion. Figure 5 shows the Winoground group score performance as a function of the image-SG pairs data portion. As can be seen, the positive slope suggests that adding image-SG data consistently improves results.

**Training with negatives from non-graph data**. To demonstrate the importance of structured information in textual descriptions, we examine the performance of the model when only LAION captions are used. Specifically, we trained using generated negatives that were not derived from SG

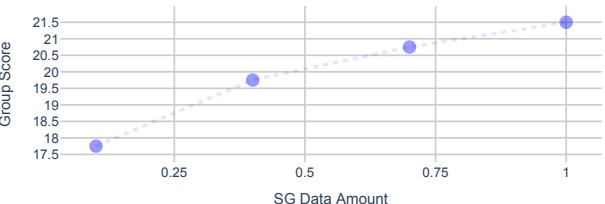

Figure 5: **Image-SG Pair Data Size**. We report the performance of our model on Winoground group score as a function of the amount of scene-graph data used during training (percentage of the available data).

data but were generated in a manner that approximated our graph-based negatives. Since we do not have the graphs in this setup, we apply the following augmentations: (i) Swapping asymmetric relations - We swap the nouns that are relevant to the relation by using a standard parser. (ii) Relation falsification - The relation is replaced with one from a closed set of relations we manually annotated in order to obtain the wrong semantic meaning. (iii) Attributes swapping - We swap attributes from a closed set of attribute categories that we manually annotated (e.g., color, etc.). The Text/Image/Group scores compared to the BLIP baseline are +4.0/-0.7/-0.8 while using BLIP with our graph-based augmentations (without SG tokens) obtains +1.5/+6.3/+4.0 compared to the BLIP baseline. It can be seen that the generated negatives from LAION improve only the Text score while applying our graph-based negatives improves all the metrics. This indicates that the main reason for the improvement is the structured information contained in the descriptions generated from the scene graphs.

**Scene graph token representations**. To analyze what the scene graph tokens learned, we can evaluate the ability of object and relationship tokens to be utilized explicitly for the auxiliary task as a simple SG predictor in images. This is accomplished by predicting the scene graphs on Visual Genome based on the learned SG tokens. We compared the learned SG tokens with a BLIP model extended with object and relationship heads, as explained in *BLIP-MT* variant in the main paper (See Section 4.5). Our model achieved an mAP of 17.7, while the *BLIP MT* achieved an mAP of 14.4. These results indicate that the scene graph tokens learn meaningful and useful representations.

**Token specialization**. Our SGVL approach learns a different specialization for each scene graph token in Figure 6. We visualize the bounding box center coordinates predicted by 10 different object

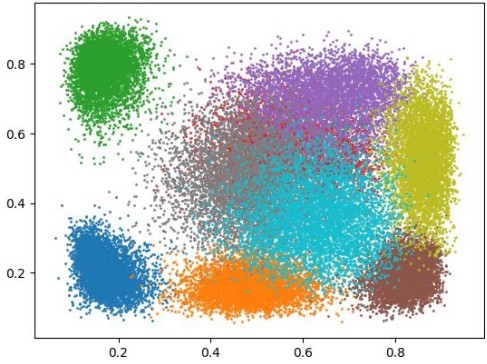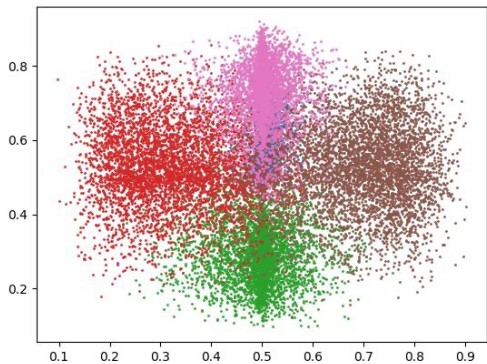

Figure 6: **Tokens Specialization**. We visualize the box predictions of 10 random object tokens (left) and 7 relationship tokens (right) on all images from the COCO validation set. Each box is represented as a point with the normalized coordinates of its center. Colors indicate the predictions made by different tokens.

| Model | Winoground | | | VL-Checklist | | | ARO | | VSR |
| | All Dataset | | | All Datasets Avg | | | Flickr30K | COCO | All Dataset |
| | Text | Image | group | Attribute | Object | Relation | Order | Order | Avg |
|---|---|---|---|---|---|---|---|---|---|
| BLIP | 39.0 | 19.2 | 15.0 | 75.2 | 82.2 | 81.5 | 27.9 | 24.9 | 56.5 |
| BLIP + Graph Text (GT) | 40.3 | 20.5 | 16.5 | 76.0 | 80.8 | 77.5 | 28.0 | 24.6 | 57.2 |
| BLIP + GT + Graph Neg. (GN) | 40.5 | 25.5 | 19.0 | 80.0 | 84.0 | 81.2 | 69.6 | 70.0 | 61.4 |
| BLIP + GT + GN + SG Tokens | 42.8 | 27.3 | 21.5 | 81.8 | 85.2 | 81.9 | 70.0 | 71.0 | 62.4 |

Table 5: Scene graph utilization ablation results for *all datasets*.

| Model | Winoground | | | VL-Checklist | | | ARO | | VSR | Graph Pred. | ZS |
| | All Dataset | | | All Datasets Avg | | | Flickr30K | COCO | All Dataset | mAP | 21 Tasks |
| | Text | Image | group | Attribute | Object | Relation | Order | Order | Avg | Score | Avg |
|---|---|---|---|---|---|---|---|---|---|---|---|
| BLIP | 39.0 | 19.2 | 15.0 | 75.2 | 82.2 | 81.5 | 27.9 | 24.9 | 56.5 | - | 49.0 |
| BLIP + SG Tokens | 39.8 | 26.5 | 20.0 | 81.0 | 84.5 | 81.2 | 67.6 | 67.0 | 61.9 | 16.1 | 47.5 |
| BLIP + Adaptive SG Tokens | 42.8 | 27.3 | 21.5 | 81.8 | 85.2 | 81.9 | 70.0 | 71.0 | 62.4 | 17.7 | 48.0 |

Table 6: Adaptive SG tokens results for *all datasets*.

tokens and 7 random relationship tokens for all images in the COCO val set. We observe that these tokens are specialized in different locations in the image, whereas the relationship tokens tend to be centered since their boxes are larger and spread over a greater area.

## A.2 Ablations on All Datasets

The main paper only includes ablation results for Winoground in Table 4a and in Table 4b. Here, in Table 5 and Table 6, we include additional ablation results. Specifically, we have performed a comprehensive ablation using our SGVL approach with the BLIP model on *all datasets*: Winoground, VL-Checklist, ARO, and VSR. It is evident from Table 5 that the utilization of our approach for both visual and textual components is important. Last, Table 6 shows that the "Adaptive SG Tokens" improve SG prediction and VL performance for all datasets

with only a mild zero-shot degradation.

## A.3 Additional Results

We start by presenting zero-shot classification results tested with linear probing. Next, we show additional results when finetuning image-text pairs from COCO in Table 7. Moreover, we show additional results on *all splits* in Winoground in Table 8, as suggested in Diwan et al. (2022). Last, we also present fine-grained results of NegCLIP, NegBLIP, and NegBLIP2 baselines on VL-Cheklist, and VSR datasets in Table 9 and Table 10. For all tables, the difference between the baselines and our SGVL approach is denoted by gains (+X) and losses (-X).

As discussed in our paper, we believe that the slight degradation in zero-shot classification is due to fine-tuning with negative captions and graph prediction tasks that deviate from the original VLM training objective. This phenomenon is also visible

| | Winoground | | | VL-Checklist | | | ARO | | VSR |
| Model | All Dataset | | | All Datasets Avg | | | Flickr30K | COCO | All Dataset |
| | Text | Image | group | Attribute | Object | Relation | Order | Order | Avg |
|---|---|---|---|---|---|---|---|---|---|
| CLIP-SGVL (COCO) | 32.8 | 11.8 | 10.0 | 73.5 | 83.3 | 81.0 | 83.0 | 79.4 | - |
| BLIP-SGVL (COCO) | 43.8 | 26.0 | 21.3 | 81.9 | 85.5 | 82.0 | 70.7 | 71.2 | 61.7 |
| BLIP2-SGVL (COCO) | 46.0 | 26.5 | 22.8 | 81.2 | 88.8 | 88.8 | 77.4 | 78.2 | 63.0 |

Table 7: **Winoground, VL-Checklist, ARO, VSR Results** when finetuning on COCO image-text pairs instead of LAION
.

in recent works (Doveh et al., 2022; Yuksekgonul et al., 2023). Here, we also report on the classification performance achieved with linear probing. For CLIP-SGVL, when using 5-shot, 10-shot linear probing, the difference on classification tasks with respect to CLIP (in the same setting) changes to -0.3%, +0.1% and with 20-shot to +0.5%. This demonstrates that partial fine-tuning does not cause degregation as previously observed in other studies.

In order to demonstrate that our method can also be used with image-text pairs other than LAION, we report in Table 7 results when used with image-text pairs from COCO (Lin et al., 2014). It can be seen that the results are comparable to our original version, which indicates the flexibility of our approach.

Table 8 shows the performance on fine-grained splits on Winoground. It can be observed that our method generally outperforms or is comparable to pretrained models for most splits. Although we present all splits, we note that Diwan et al. (2022) suggested that only the samples from the NoTag split are actually probing compositionality, while the other splits are difficult to solve for various additional reasons. Thus, the NoTag is the most important split to evaluate.

Table 9 shows the performance on fine-grained splits of the VL-Checklist dataset, including the "Attribute", "Object", and "Relation" splits. As can be seen, our method improves both the pretrained baselines CLIP, BLIP, and BLIP2, as well as the NegCLIP, NegBLIP, and NegBLIP2 baselines on those fine-grained splits.

Table 10 shows the performance on fine-grained splits of the VSR dataset, which test spatial understanding in VLMs. Similarly, we find that our method improves on most splits and across models when compared with pretrained and Neg baselines.

Overall, our approach improves multiple architectures (CLIP, BLIP, and BLIP2) on a variety of VL datasets with only mild degradation in zero-shot performance.

## B Additional Modeling Details

We begin by presenting additional model details regarding our graph preprocessing procedure (Section B.1). Next, we describe our method for creating graph-based negatives (Section B.2), which is illustrated in Figure 7. We provide more details on our scene graph loss (Section B.3) and some modifications made to our loss calculation when training BLIP (Li et al., 2022b) and BLIP2 (Li et al., 2023) (Section B.4). We conclude by describing more in detail our approach when using the BLIP2 model (Section B.5).

### B.1 Graph Preprocessing

We next describe how we process the image-SG pairs to create our training dataset. Our guiding principle is to create image-SG training pairs where the graphs are dense enough but not too large, in order to allow structured and short descriptions. To this end, given an image $I$ and a corresponding graph $G(V, E)$, we extract the sub-graphs by taking a random walk on the graph. The random walk is initialized by randomly picking a relationship from the graph (edge $e \in E$ and nodes $v_1, v_2 \in V$ such that $e = (v_1, v_2)$) and ends when a node that has no outgoing edges is reached, resulting in a sub-graph $G_1 = (V_1, E_1)$. Next, the image is cropped to the union of the bounding boxes of all objects ($v \in V_1$) in the extracted sub-graph, resulting image $I_1$. We finish the process by adding new nodes and relationships to $G_1$ from the residual graph $G_r = (V/V_1, E/E_1)$ that are visible in $I_1$. We use $G_1$ and $I_1$ as a training sample only if the derived $G_1$ contains at most 10 objects (i.e. $|E_1| \leq 10$). This process creates SGs composed of connected components that are all DAGs with a single Hamiltonian path, which facilitates caption generation.

### B.2 Graph-Based Negatives

In order to generate negative image captions, we propose a set of predefined rules that when applied to an image-scene graph pair, result in a

| Model | Ambiguosly Correct | | | Unusual Image | | | Unusual Text | | | Non Compositional | | | Visually Difficult | | | Complex Reasoning | | | No Tag | | |
|---|---|---|---|---|---|---|---|---|---|---|---|---|---|---|---|---|---|---|---|---|---|
| | T | I | G | T | I | G | T | I | G | T | I | G | T | I | G | T | I | G | T | I | G |
| CLIP | 30.4 | 15.2 | 13.0 | 25.0 | 8.9 | 5.4 | 30.0 | 16.0 | 10.0 | 76.7 | 36.7 | 33.3 | 15.8 | 0.0 | 0.0 | 24.4 | 7.7 | 3.8 | 30.4 | 11.1 | 8.2 |
| BLIP | 39.1 | 17.4 | 15.2 | 37.5 | 16.1 | 14.3 | 30.0 | 14.0 | 8.0 | 50.0 | 33.3 | 30.0 | 29.0 | 10.5 | 10.5 | 24.4 | 7.7 | 2.6 | 44.8 | 23.8 | 19.2 |
| BLIP2 | 41.3 | 28.3 | 19.6 | 30.4 | 17.9 | 14.3 | 38.0 | 14.0 | 12.0 | 53.3 | 26.7 | 26.7 | 34.2 | 13.1 | 10.5 | 20.5 | 7.6 | 2.5 | 50.0 | 32.0 | 26.7 |
| NegCLIP | 28.2 | 4.3 | 4.3 | 17.9 | 7.2 | 3.6 | 36.0 | 8.0 | 8.0 | 66.7 | 30.0 | 26.7 | 10.5 | 2.6 | 2.6 | 21.8 | 7.7 | 5.1 | 33.2 | 12.2 | 8.7 |
| NegBLIP | 43.5 | 21.7 | 8.7 | 42.9 | 19.6 | 14.3 | 28.0 | 18.0 | 12.0 | 56.7 | 43.3 | 33.3 | 28.9 | 15.8 | 10.5 | 32 | 10.2 | 3.8 | 47.0 | 27.3 | 22.6 |
| NegBLIP2 | 41.3 | 23.9 | 19.6 | 35.7 | 17.8 | 17.8 | 36.0 | 14.0 | 14.0 | 63.3 | 33.3 | 30.0 | 23.7 | 15.8 | 13.1 | 24.3 | 9.0 | 6.4 | 49.4 | 34.3 | 27.9 |
| CLIP-SGVL (ours) | 37.0 | 13.0 | 8.7 | 28.6 | 10.7 | 7.1 | 32.0 | 10.0 | 8.0 | 70.0 | 40.0 | 30.0 | 18.4 | 5.3 | 5.3 | 24.4 | 16.7 | 12.8 | 33.2 | 14.0 | 8.7 |
| BLIP-SGVL (ours) | 45.6 | 21.7 | 21.7 | 44.6 | 23.2 | 21.4 | 38.0 | 24.0 | 18.0 | 46.7 | 43.3 | 40.0 | 23.7 | 18.3 | 15.8 | 33.3 | 11.2 | 7.7 | 45.9 | 34.3 | 25.6 |
| BLIP2-SGVL (ours) | 43.5 | 26.1 | 23.9 | 39.3 | 26.8 | 21.4 | 34.0 | 18.0 | 16.0 | 60.0 | 46.7 | 46.7 | 26.3 | 18.4 | 15.3 | 29.5 | 11.5 | 6.4 | 51.7 | 37.2 | 29.0 |

Table 8: **Results on Winoground** for all the splits presented in Diwan et al. (2022). We report T, I, and G for Text retrieval, Image retrieval, and Group retrieval.

| Model | Attribute Action | Color | Material | Size | State | Object Location | Size | Relation Action |
|---|---|---|---|---|---|---|---|---|
| CLIP | 68.1 | 70.2 | 73.1 | 52.9 | 63.3 | 81.0 | 80.1 | 78.0 |
| BLIP | 79.5 | 83.2 | 84.7 | 59.8 | 68.8 | 83.0 | 81.3 | 81.5 |
| BLIP2 | 81.0 | 86.2 | 90.3 | 61.7 | 70.1 | 85.4 | 84.3 | 84.9 |
| NegCLIP | 66.7 | 74.9 | 78.4 | 54.8 | 63 | 81.9 | 80.9 | 81.3 |
| NegBLIP | 82.6 | 88.3 | 89.3 | 60.8 | 70.1 | 81.0 | 79.6 | 83.0 |
| NegBLIP2 | 82.1 | 88.7 | 90.7 | 63.0 | 70.0 | 87.2 | 86.8 | 88.2 |
| CLIP-SGVL (ours) | 76.6 (+8.5) | 78.7 (+8.5) | 81.3 (+8.2) | 59.7 (+6.8) | 62.0 (-1.3) | 83.2 (+2.2) | 82.0 (+1.9) | 81.3 (+3.3) |
| BLIP-SGVL (ours) | 79.2 (-0.3) | 94.5 (+11.3) | 91.9 (+7.2) | 73.3 (+13.5) | 70.0 (+1.2) | 86.4 (+3.4) | 83.9 (+2.6) | 81.9 (+0.4) |
| BLIP2-SGVL (ours) | 82.4 (+1.4) | 91.7 (+5.5) | 92.2 (+1.9) | 70.1 (+8.4) | 69.6 (-0.5) | 89.0 (+4.6) | 87.6 (+3.3) | 88.8 (+3.9) |

Table 9: **VL-Checklist Results**. We report VL-Checklist (Zhao et al., 2022) on all splits of the Attribute, Object, and Relation tests, excluding the Visual Genome dataset. The difference between base models and SGVL is denoted by (+X).

| Model | Adjacency | Directional | Orientation | Projective | Proximity | Topological | Unallocated | Average |
|---|---|---|---|---|---|---|---|---|
| BLIP | 55.4 | 48.9 | 53.7 | 58.2 | 56.5 | 55.6 | 63.4 | 56.5 |
| BLIP2 | 56.2 | 47.9 | 59.8 | 62.5 | 55.8 | 66.7 | 66.3 | 61.9 |
| NegBLIP | 56.0 | 48.0 | 55.7 | 59.0 | 55.6 | 58.7 | 63.2 | 57.8 |
| NegBLIP2 | 56.3 | 48.0 | 57.4 | 61.5 | 55.8 | 67.8 | 70.5 | 61.2 |
| BLIP-SGVL (ours) | 57.7 (+2.3) | 54.1 (+5.2) | 57.8 (+4.1) | 63.8 (+5.6) | 57.8 (+1.3) | 64.8 (+8.8) | 68.8 (+5.4) | 62.4 (+5.9) |
| BLIP2-SGVL (ours) | 59.8 (+3.6) | 56.9 (+9.0) | 58.5 (-1.3) | 61.5 (-1.0) | 59.7 (+3.9) | 70.0 (+3.3) | 66.8 (+0.5) | 63.4 (+1.5) |

Table 10: **VSR Results.** We report accuracy on all splits of the VSR (Liu et al., 2022) dataset.

negative scene graph that incorrectly describes the image. Our scene graph to caption scheme transforms negative scene graphs into negative captions that are semantically inconsistent with the images they accompany. For each training sample we randomly apply one of the following negative rules focused on object attributes and relationships in the graph: (i) *asymmetric relations swapping* - we call a relationship $R$ asymmetric, if for two objects $a, b$, $aRb \implies \neg bRa$. We manually annotated a relation as asymmetric out of the 300 most common VG relations. We use these to generate a negative scene graph by searching for an edge $e = (v_1, v_2) \in E$ representing an asymmetric relationship, and modify the graph by replacing $e$ with an edge $e_n = (v_2, v_1)$. For example, in the case of a graph describing the phrase "dog chasing cat", such a negative will result in the phrase "cat chasing dog". (ii) *relation falsification* - we replace relations in the graph with false relations. For example, we turn "cup on table" to "cup under table". For negatives focused on object attributes, we first scan the dataset and split the attributes into the following categories: *color*, *material*, *size*, *state*. Next, we use this split to perform two types of negatives: (i) *attributes falsification* - we replace attributes for objects in the graph with false attribute from the same category. For example, turning "blue ball" to "red ball". (ii) *attributes swapping* - we search the graph for two objects that are annotated with attributes from the same category. Given that such a pair has been found we switch between the attributes, result-

ing for example, "silver spoon and golden knife" from "golden spoon and silver knife".

## B.3  Scene Graph Loss

This section provides a detailed explanation of our Scene Graph loss, as mentioned in the main paper in Section 3.4.

As explained in the paper (See Section 3.4), we incorporate SG tokens into the model, which are used to predict the SG that corresponds to the image. We next explain this process. The graph $G$ contains several annotations: the set of object categories $O = \{o_i\}_{i=1}^n$, the set of bounding boxes $B^O = \{b_i^o\}_{i=1}^n$, and set of relationship categories $R = \{r_i\}_{i=1}^m$. We also augment the relationships with a set of bounding boxes, that are constructed as the union of the boxes of the corresponding objects, and denote these by $B^R = \{b_i^r\}_{i=1}^m$.

As we use an open vocabulary approach, we do not aim to predict the object and relationship categories directly, but rather use the embeddings from the VL model. Thus, we extract the embeddings of these labels with the text encoder $E_T$ to get class embeddings: $\tilde{O} = E_T(O) \in \mathbb{R}^{(n+1)\times d}$, and $\tilde{R} = E_T(R) \in \mathbb{R}^{(m+1)\times d}$. We note that $(n+1)$ and $(m+1)$ classes are due to "no object" and "no relationship" categories (denoted as $\varnothing$), which are represented with learned embeddings.

Thus far we described the elements of the SG that we aim to predict from the SG tokens. We next describe the prediction process, and the corresponding losses. The image encoder outputs a set of $\tilde{n}$ object queries and $\tilde{m}$ relationship queries. We apply two feed-forward networks, $\text{FFN}_{bb}$ and $\text{FFN}_e$, on top of these queries to predict bounding boxes and label embeddings respectively. Specifically, given the final representations of $j^{th}$ object token and $k^{th}$ relationship token for image $I$, which we denote $F_O^j(I)$ and $F_R^k(I)$ respectively, we predict:

$$\hat{b}_j = \text{FFN}_{bb}(F_O^j(I)) \ , \ \hat{e}_j = \text{FFN}_e(F_O^j(I)) \quad (7)$$

$$\hat{b}_k = \text{FFN}_{bb}(F_R^k(I)) \ , \ \hat{e}_k = \text{FFN}_e(F_R^k(I)) \quad (8)$$

where $\hat{b}_j$, $\hat{b}_k \in \mathbb{R}^{1\times 4}$ are bounding box predictions and $\hat{e}_j$, $\hat{e}_k \in \mathbb{R}^{1\times d}$ are class embeddings predictions. Next, we use the class embeddings matrices to predict probabilites over $n+1$ and $m+1$ classes:

$$\hat{q}_j^o = \text{SoftMax}(\hat{e}_j \tilde{O}^T) \quad (9)$$

$$\hat{q}_k^r = \text{SoftMax}(\hat{e}_k \tilde{R}^T) \quad (10)$$

where $\hat{q}_j^o \in \mathbb{R}^{1\times n+1}$ and $\hat{q}_k^r \in \mathbb{R}^{1\times m+1}$. Next, we need to match the predictions of the SG tokens with the ground-truth SG, in order to determine which SG tokens correspond to which ground-truth objects and relationships. We follow the matching approach as in DETR (Carion et al., 2020), except that in our case, objects and relationships are matched separately. We describe the object matching below. Given a permutation $\sigma$ over the object tokens, we define the matching-cost between the permutation and the GT by:

$$s(\sigma) = \sum_{i=1}^{\tilde{n}} \Big[ \mathbb{1}_{\{o_i \neq \varnothing\}} \hat{q}_{\sigma(i)}^o(o_i) +$$
$$\mathbb{1}_{\{o_i \neq \varnothing\}} \Big( \mathcal{L}_{giou}(b_i^o, \hat{b}_{\sigma(i)}) + \|b_i^o - \hat{b}_{\sigma(i)}\|_1 \Big) \Big] (11)$$

Namely, we check the compatibility between the GT and permuted objects both in terms of object category (i.e., the probability assigned by the query to the GT object $o_i$) and in terms of bounding boxes (i.e., how well the predicted box matches the GT one). Here $\mathcal{L}_{giou}$ is from (Rezatofighi et al., 2019).

The optimal matching $\hat{\sigma}$ is found by optimizing this score: $\hat{\sigma} = \arg\min_{\sigma \in \Sigma} s(\sigma)$. Finally, we use the optimal matching $\hat{\sigma}$ from above to calculate the following Objects loss:

$$\mathcal{L}_{Objects} = \sum_{i=1}^{\tilde{n}} \Big[ -\log \hat{q}_{\hat{\sigma}(i)}^o(o_i) +$$
$$\mathbb{1}_{\{o_i \neq \varnothing\}} \Big( \mathcal{L}_{giou}(b_i^o, \hat{b}_{\hat{\sigma}(i)}) + \|b_i^o - \hat{b}_{\hat{\sigma}(i)}\|_1 \Big) \Big] (12)$$

The relation matching and loss $\mathcal{L}_{Rel}$ is calculated in a similar manner, and the total scene graph loss is the sum of $\mathcal{L}_{Obj}$ and $\mathcal{L}_{Rel}$:

$$\mathcal{L}_{SG} := \mathcal{L}_{Obj} + \mathcal{L}_{Rel} \quad (13)$$

## B.4  BLIP Image-Text Loss Details

Besides image and text unimodal encoders trained using a contrastive loss, BLIP (Li et al., 2022b) and BLIP2 (Li et al., 2023) also includes an image-grounded text encoder that uses additional image-text cross-attention layers. The encoder is equipped with a binary classification head (a linear layer) and is trained to predict whether an image-text pair is positive (matching) or negative (unmatching). In the training procedure described by the authors, the encoder uses a hard negative mining strategy to calculate an additional loss for all image-text pairs in the batch. When training our BLIP/BLIP2-SGVL models, we apply this loss as well. Additionally,

we use this encoder to add another term to our graph-based negative loss ($\mathcal{L}_{GN}$). Let $E_{IT}^P(I,T)$ denote the positive score given by the encoder to some image-text pair $(I,T)$, then the following term is added to $\mathcal{L}_{GN}$:

$$\sum_{I_G,T_P,T_N} -log\left(\frac{e^{E_{IT}^P(I_G,T_P)}}{e^{E_{IT}^P(I_G,T_P)} + e^{E_{IT}^P(I_G,T_N)}}\right) \quad (14)$$

with $I_G, T_P, T_N$ denoting the VG images, positive and negative captions in the batch, respectively.

### B.5 BLIP2 Model Details

In this section, we describe how we incorporate our "Adaptive SG Tokens" into the BLIP2 (Li et al., 2023) model. Recall that the BLIP2 model architecture is based on a Q-Former module that consists of two transformer sub-modules: (a) an image transformer that interacts with a frozen image encoder to produce the visual features used for contrastive learning; and (b) a text transformer that performs both the functions of a text encoder (producing the textual features required for contrastive learning) and a text decoder. The inputs to the image transformer sub-module are a set of learnable query embeddings. These queries interact with each other through self-attention layers, as well as with frozen image encoder features through cross-attention layers. When applying our SGVL approach to BLIP2, we add our SG tokens as an additional set of prompts in parallel to the learnable queries. Therefore, our SG tokens interact with the learnable queries and the frozen image encoder features. We apply our adaptation technique to these tokens as we do in BLIP and CLIP (see Adaptive SG Tokens in section 3.3) and use the exact same procedure for the graph prediction task.

### C Qualitative Visualizations

Figure 7 shows a visualization of the generation process of captions, including positive captions as well as negative captions, based on our Graph-based Negatives module. As shown in the figure, captions generated from scene graphs are much more focused on describing fine-grained details. Furthermore, we show in Figure 8 visualizations of scene graph tokens predictions for images from Visual Genome, which the model was not trained on. It can be seen that although the model has not been trained on these images, the predictions are reasonable. Finally, we show in Figure 9 and Figure 10

error analysis on Winoground and VL-Checklist to evaluate the success and errors of our method and the baselines. This illustrates which examples our BLIP-SGVL model is successful on, in contrast to the BLIP model.

### D Additional Implementation Details

Our SGVL approach can be used on top of a variety of VL models. For our experiments, we choose the CLIP (Radford et al., 2021), BLIP (Li et al., 2022b) and BLIP2 (Li et al., 2023) models as they are among the most popular and easy-to-use methods. These models are implemented based on the Open-CLIP library (Ilharco et al., 2021) and the BLIP/BLIP2 code base (available at https://github.com/salesforce/LAVIS). We implement SGVL based on these repositories. As described above, our approach is trained using both the original image-text pairs from LAION and the image-SG pairs we curate from Visual Genome. In particular, for CLIP-SGVL we use 3M image-text pairs, while for BLIP/BLIP2-SGVL, we use 750K due to computational constrains.

In our experiments, we trained our CLIP-SGVL on 4 V100 GPUs for 32 epochs with a batch comprised of 256 image-text pairs and 8 image-SG pairs. We use AdamW optimizer (Kingma and Ba, 2014; Loshchilov and Hutter, 2017) with $\beta_1 = 0.9$, $\beta_2 = 0.98$, and $\epsilon = 1e-6$. We use $lr = 5e-5$ with cosine scheduler, and a weight decay of $0.2$ for regularization. For BLIP-SGVL, we trained on 4 V100 GPUs and for BLIP2-SGVL on 4 A100 GPUs. We train both for 8 epochs with a batch comprised of 32 image-text pairs and 8 image-SG pairs. We use AdamW optimizer (Kingma and Ba, 2014) with $\beta_1 = 0.9$, $\beta_2 = 0.99$, and $\epsilon = 1e-8$. We use $lr = 5e-5$ with cosine scheduler, and a weight decay of $0.05$ for regularization.

### D.1 VL-Checklist

**Dataset**. VL-Checklist (Zhao et al., 2022) is a new study that combines the following datasets: Visual Genome (Krishna et al., 2017), SWiG (Pratt et al., 2020), VAW (Pham et al., 2021), and HAKE (Li et al., 2019). For each image, two captions are given, a positive and a negative. The positive caption is derived from the source dataset and is coherent with the visual structure of the image. the negative caption is constructed by modifying one word in the positive caption that corresponds to a structural aspect in the image. To correctly solve

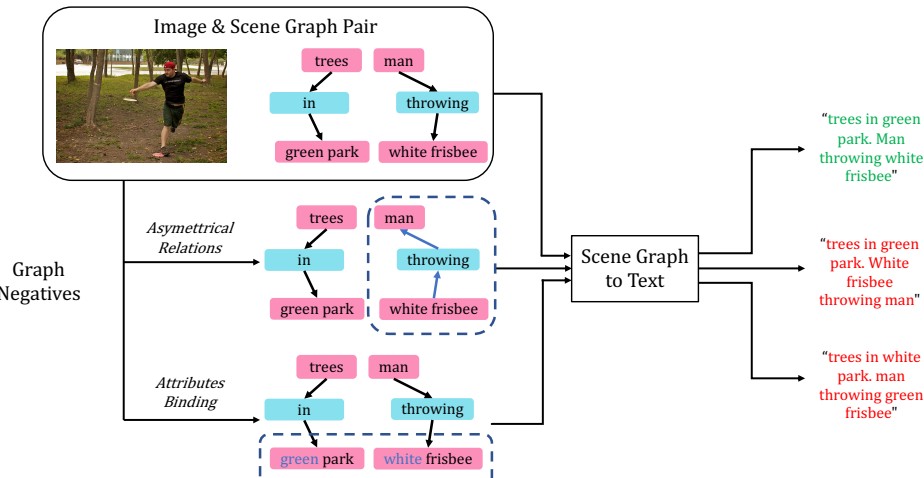

Figure 7: **Visualization** of some of our Graph-based Negatives as well as the SG-to-Text module. We show the generation process of positive captions (green) and negative captions using the graph (red).

a sample the model needs to identify the caption faithfully describing the image. Specifically, VL-Checklist evaluates the following structured concepts: (1) Object: identifying whether objects mentioned in the text appear in the image invariantly to their spatial location and size, (2) Relation: spatial or action relation between two objects, and (3) Attribute: color, material, size, state, and action bounded to objects. We report results on a combined VL-Checklist dataset excluding VG.

**Inference details**. We use the official data and code released by the authors which is available at `https://github.com/om-ai-lab/VL-CheckList`. A test sample consists of an image and two captions. For CLIP, we compute the cosine similarity between the image and the captions and report the positive caption as the one with the higher similarity. For BLIP/BLIP2 we use the ITM head, which predicts both a positive and negative score for each pair. We consider the caption with the higher positive score to be the correct one.

### D.2 Winoground

**Dataset**. Winoground (Thrush et al., 2022) is a new challenging dataset that evaluates the ability of VL models to capture compositionality in vision & language. The dataset contains 1600 tests across 400 samples. Each sample is composed of two image-text pairs $(I_0, C_0), (I_1, C_1)$. The pairs have overlapping lexical content but are differentiated by a swapping of an object, a relation, or both. To correctly solve the sample the model needs to correctly solve two text retrieval and two image

retrieval tasks. A recent study (Diwan et al., 2022) has shown that solving Winoground requires not just compositional understanding but also other abilities such as commonsense reasoning. The study proposed a new split to the dataset differentiating the samples by the source of their hardness. Specifically, the split of the samples into the following categories is as follows: *Non Compositional* - There are 30 samples in this category that do not require compositional reasoning. *Visually Difficult* - The model must be able to detect an item that is visually difficult to identify (small, blurry, in the background, etc.) in order to sort these samples correctly. This category includes 38 samples. *Ambiguously Correct* - This category includes 46 samples where at least one caption accurately describes both images or doesn't quite describe any of the images. *Unusual Text & Unusual Image* - There are 106 samples in these categories, all of which contain unrealistic or awkward texts or images that make it difficult to solve them with a VL model. *Complex Reasoning* - This category consists of 78 samples that require common sense reasoning or knowledge of the world around us. *No Tag* - These are vanilla Winoground examples that solely probe compositional understanding.

**Inference details**. We use the official data and code released by the authors which is available at `https://huggingface.co/datasets/facebook/winoground`. For testing, the pairs are given, and a text score, an image score, and a group score for a sample is computed in the following way: The text score is 1 if and only if image $I_0$

has a higher similarity to caption $C_0$ than $C_1$, and image $I_1$ has a higher similarity to caption $C_1$ than $C_0$. Similarly the image score is 1 if and only if caption $C_0$ has a higher similarity to image $I_0$ than image $I_1$ and $C_1$ has a higher similarity to image $I_1$ than image $I_0$. The group score is 1 if and only if both text and image scores are 1. Thus, the random chances for both the image and text score, is $1/4$ while for group score it is $1/6$. Similarities between image-text pairs is computed as in section D.1.

### D.3 ARO

**Dataset**. ARO (Yuksekgonul et al., 2023) (Attribution, Relation, and Order) is a new benchmark that tests compositionality in VL models. The authors propose four tasks that are sensitive to order and composition, namely Visual Genome Relation, Visual Genome Attribution, COCO& Flickr30k Order. Since our approach is trained on Visual Genome, we report only the COCO and Flickr30k order task (PRC). For the order task, image-text pairs from the mentioned datasets are used. The words in the text are reordered in order to create false captions for the image, according to the following perturbations: *nouns and adjectives shuffle*, *everything but nouns and adjectives shuffle*, *trigrams shuffle* and *words within trigrams shuffle*.

**Inference details**. We use the official data and code released by the authors which is available at `https://github.com/mertyg/vision-language-models-are-bows`. During inference, each sample consists of an image and five textual descriptions. The similarity of each text to the image is measured as in section D.1, and the text with the highest similarity to the image is reported as the real caption.

### D.4 VSR

**Dataset**. VSR (Liu et al., 2022) VSR (Visual Spatial Reasoning) is a new benchmark for measuring the spatial understanding of vision-language models. The VSR dataset consists of natural image-text pairs in English, each example contains an image and a natural language description of the spatial relationship between two objects shown in the image. The VL model needs to classify images and captions as either true or false, indicating whether a caption accurately describes the spatial relationship. The dataset has more than $10K$ image-text samples, derived from 6,940 COCO images and covers 65 spatial relations. The dataset is split into a train, validation and test sets, however, since we

evaluate in a zero-shot manner we test our model and baselines using all samples from the train, validation, and test splits. The spatial relations are divided into 7 meta-categories: *Adjacency*, *Directional*, *Orientation*, *Projctive*, *Proximity*, *Topological*, *Unallocated*. We report results according to these categories, as well as the average over all spatial relations.

**Inference details**. We use the official data released by the authors which is available at `https://github.com/cambridgeltl/visual-spatial-reasoning` We do not evaluate CLIP on this task since the task requires assigning a true or false label to an image-text pair. CLIP, however, does not allow this to be done in a straightforward manner, and Therefore only the BLIP/BLIP2 models can be used. We use the ITM head to determine whether the sample is true or false.

### D.5 Finetuning Datasets

In our work, we use the LAION dataset as "standard" image-text pairs, along with image-SG data pair from Visual Genome (Krishna et al., 2017) (VG). Visual Genome is annotated with $108,077$ images accompanied by their corresponding scene graphs. On average, images have 35 entities, 21 relationships, and 26 attributes per image. Additionally, there are approximately 70K object categories and 40K relationship categories. In general, Visual Genome scene graphs can be viewed as dense knowledge representations for images, similar to the format used for knowledge bases in natural language processing.

### D.6 Licenses and Privacy

The license, PII, and consent details of each dataset are in the respective papers. In addition, we wish to emphasize that the datasets we use do not contain any harmful or offensive content, as many other papers in the field also use them. Thus, we do not anticipate a specific negative impact, but, as with any Machine Learning method, we recommend to exercise caution.

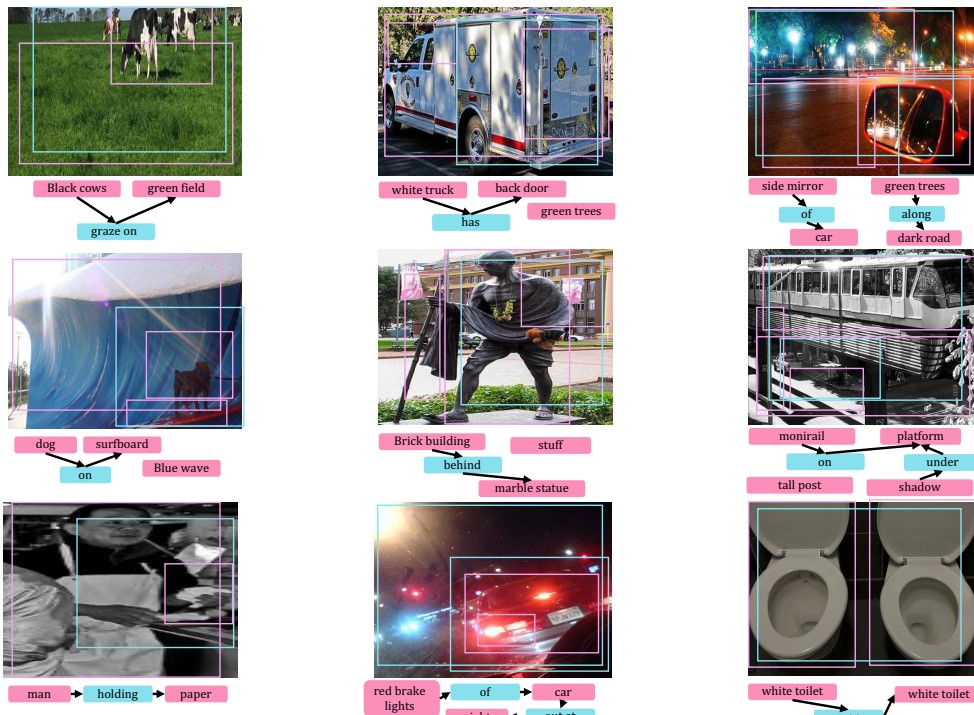

Figure 8: **Scene Graph Prediction**. We show the predictions of the "scene graph tokens" on images from Visual Genome that were not trained by our model.

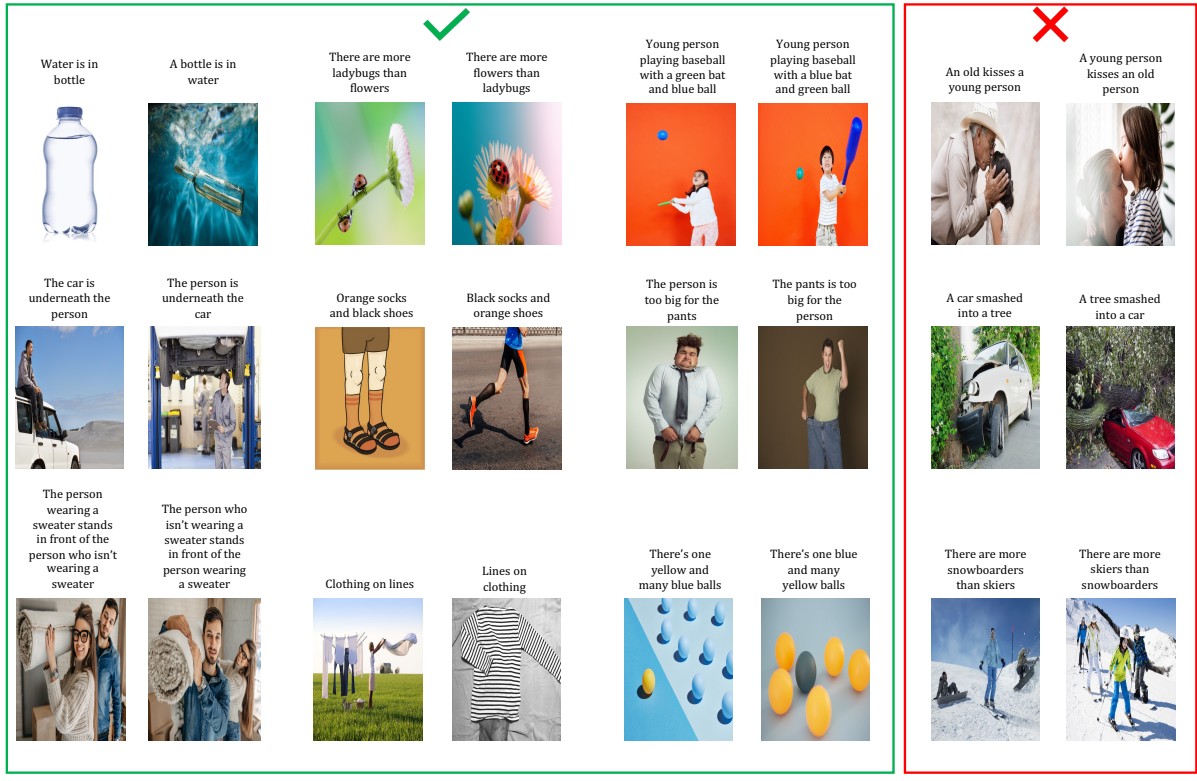

Figure 9: **Error Analysis on Winoground**. We demonstrate on the left in green where our BLIP-SGVL model succeeds, while the baseline BLIP model fails. On the right, in red, we can observe examples in which our BLIP-SGVL model fails. As visible, our model improves in samples that require understanding relations between objects, binding attributes to objects, and counting objects.

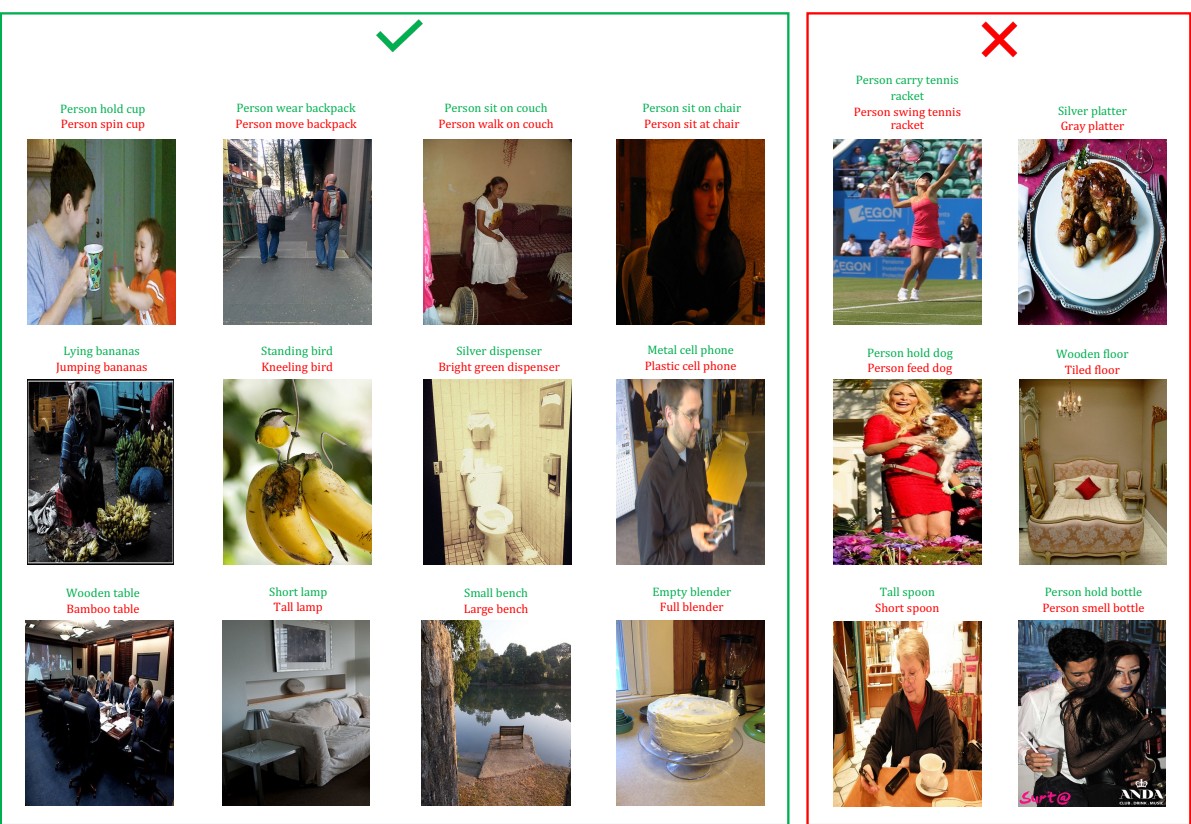

Figure 10: **Error Analysis on VL-Checklist**. We demonstrate on the left in green where our BLIP-SGVL model succeeds, while the baseline BLIP model fails. On the right, in red, we can observe examples in which our BLIP-SGVL model fails. True captions are in **green** and false captions in **red**. As visible, some of the samples on which our model fails are ambiguous or visually difficult to solve.