# OpenReview forum: "Incorporating Structured Representations into Pretrained Vision \& Language Models Using Scene Graphs"
_EMNLP/2023/Conference — EMNLP 2023 Main_

### Official Review · Reviewer_VrYK · 2023-08-03

**Soundness:** 3

**Excitement:**

3: Ambivalent: It has merits (e.g., it reports state-of-the-art results, the idea is nice), but there are key weaknesses (e.g., it describes incremental work), and it can significantly benefit from another round of revision. However, I won't object to accepting it if my co-reviewers champion it.

**Paper Topic And Main Contributions:**

The work proposes to teach vision-language Transformers structured information leveraging scene graphs (SGs). Supervision signals are obtained from SGs in two ways: 1)  positive and negative captions are generated with different compositional substituents; 2) object labels, relations and coordinates are used to supervise a set of special "scene graph tokens".

The learnable scene graph tokens possess a separate set of attention and feedforward-layer parameters. They are allowed to interact with regular patch tokens through self-attention, which can be later leveraged for predicting the text embedding of objects or relations.

The entire model is finetuned from CLIP/BLIP via LoRA and outperforms various baselines in hard VL benchmarks requiring compositional scene understanding.


**Questions For The Authors:**

Line 263: How do you represent locations for relationship tokens?

**Reasons To Accept:**

This paper demonstrates that the standard contrastive language-image pretraining cannot sufficiently imbue a model with compositional understanding capabilities. Therefore, the authors propose to leverage information about relations and attributes from scene graphs. It's demonstrated that a small number of scene graph annotations could successfully compensate for the lack of compositional understanding. The results would encourage the community to adapt models for better compositional scene understanding with moderate effort.

**Reasons To Reject:**

The proposed method does not consistently improve performances. As mentioned in Line 485-496, zero-shot performance was degraded. Table 2&3 also show that the  proposed method harmed performances on certain partitions of the evaluation benchmark. I'm expecting more error analysis on the cause of such degradation. For example, is there a commonality of all tasks where performances were harmed by SGVL, such that during application, users could wisely choose when to adopt BLIP vs. BLIP-SGVL?

**Reproducibility:**

4: Could mostly reproduce the results, but there may be some variation because of sample variance or minor variations in their interpretation of the protocol or method.

**Reviewer Confidence:**

4: Quite sure. I tried to check the important points carefully. It's unlikely, though conceivable, that I missed something that should affect my ratings.

**Typos Grammar Style And Presentation Improvements:**

Line603 qualitative annotations --> high-quality annotations

Line605-606: I don't think unsupervised training is a natural extension of this work. The main insight brought by this work is the necessity of a small amount of densely annotated data. Future directions might include generalizing this approach beyond the VL domain or using other types of dense structured data (e.g. segmentation masks, sketch)

---

> ### Author Rebuttal · Authors · 2023-08-27
>
> ### Shared Comments
>
> We thank the reviewers for their insightful comments. We are encouraged that they found the proposed idea of leveraging small scene graph datasets **could benefit the research community immensely** `(dPsS, VrYK)` and **could successfully compensate for the lack of compositional understanding** `(VrYK)`. Furthermore, they found the proposed framework **reasonable and effective** `(639u)` and the architectural changes and losses **easy to implement** `(dPsS)`. Finally, they observed **comprehensive ablations** `(639u, dPsS)`, demonstrating **significant improvements** over the proposed tasks `(639u)`. We are pleased that the reviewers found the paper **well written** `(639u, dPsS)`, and the **diagrams capture** the novelty aspect.
>
> A brief summary of the main aspects of our rebuttal response:
> 1) We performed additional experimental comparisons to other methods, including SOTA VLMs `(639u)`.
>
> 2) We discussed the comparison between our work and other related works and explained in more detail our novelty compared to other works `(639u)`.
>
> 3) We performed a comprehensive analysis of the degradation in performance in certain partitions of the evaluated benchmark and the zero-shot degradation `(VrYK)`.
>
> 4) We performed an experiment replacing 1% of LAION with CC3M for training `(dPsS)`.
>
> We next address all concerns and look forward to an open and constructive discussion with the reviewers.
>
>
> ### Response to VrYK
>
>
> We thank the reviewer for the insightful comments. In the following, we provide a response to the questions raised in the review:
>
> Error analysis. Thank you for your suggestion. We performed error analysis on the cause of degradation in performance in the evaluated subsets below.
>
> For Table 2, we believe that the minor degradation in state and action splits of VL Checklist may be the result of our generated negative captions being less focused on these aspects. Following your comment, we designed a model with additional negatives such that more attributes of this kind (action, state) are replaced with opposite attributes during finetuning. We note that we only included the 300 most common attributes in our submission. In order to allow more action attributes and state attributes to be replaced, we increased this number to 500. We report again performance on VL-Checklist *Attribute* splits for the base models (CLIP, BLIP, BLIP2), our models from the submission (CLIP-SGVL, BLIP-SGVL, BLIP2-SGVL), and the new mentioned models (CLIP-SGVL^, BLIP-SGVL^, BLIP2-SGVL^). It can be seen that this mitigates the degradation for these splits.:
>
>
> |             | Action       | Color        | Material     | Size         | State        |
> |-------------|--------------|--------------|--------------|--------------|--------------|
> |     CLIP    |     68.1     |     70.2     |     73.1     |     52.9     |     63.3     |
> |     BLIP    |     79.5     |     83.2     |     84.7     |     59.8     |     68.8     |
> |    BLIP2    |     81.0     |     86.2     |     90.3     |     61.7     |     70.1     |
> |  CLIP-SGVL (Ours)  |  76.6 (+8.5) | 78.7  (+8.5) | 81.3  (+8.2) | 59.7  (+6.8) | 62.0  (-1.3) |
> |  BLIP-SGVL (Ours)  | 79.2  (-0.3) | 94.5 (+11.3) | 91.9  (+7.2) | 73.3 (+13.5) | 70.0  (+1.2) |
> |  BLIP2-SGVL (Ours) |  82.4 (+1.4) | 91.7  (+5.5) | 92.2  (+1.9) | 70.1  (+8.4) | 69.6  (-0.5) |
> |  CLIP-SGVL^ (Ours) | 78.1 (+10.0) | 78.5  (+8.3) | 81.5  (+8.4) | 59.7  (+6.8) | 65.0  (+1.7) |
> |  BLIP-SGVL^ (Ours) |  80.5 (+1.0) | 94.5 (+11.3) | 91.7  (+7.0) | 73.5 (+13.7) | 71.4  (+2.6) |
> | BLIP2-SGVL^ (Ours) |  83.5 (+2.5) | 91.7  (+5.5) | 92.4  (+2.1) | 69.8  (+8.1) | 72.0  (+1.9) |
>
> We note that the rest of the results were not affected, and therefore, we have not included them in this analysis.
>
> For Table 3 in the submission, there is a minor degradation in the performance of BLIP2-SGVL on the *orientation* and *projective* splits of VSR. We note that these two splits include spatial relations such as: “left of”, “right of”, “in front of”. As mentioned in the VSR paper (see Figures 1 and 2 in [1]), interpreting these spatial relations requires choosing a frame of reference. For some images, a statement can be both true and false, depending on this choice. We compare the results of BLIP2 and BLIP2-SGVL per spatial relation for these two subsets (see table below). The results show that degradation is indeed specific to these relations. We, therefore, hypothesize that some VG annotations might have a different interpretation of these relations, and therefore fine-tuning results in degradation.
>
>
> |            | On top of   | beneath     | beside      | behind      | Left of     | Right of    | under       | In front of | below       | above       | over        | In the middle of |
> |------------|-------------|-------------|-------------|-------------|-------------|-------------|-------------|-------------|-------------|-------------|-------------|------------------|
> | BLIP2      | 74.5        | 61.2        | 67.6        | 64.2        | 51.3        | 51.5        | 59.8        | 63.5        | 59.1        | 54.1        | 55.4        | 60.0             |
> | BLIP2-SGVL (Ours) | 75.5 (+1.0) | 63.5 (+2.3) | 70.5 (+2.9) | 66.2 (+2.0) | 47.5 (-3.8) | 45.5 (-6.0) | 62.8 (+3.0) | 60.0 (-3.5) | 60.0 (+0.9) | 55.4 (+1.3) | 56.0 (+0.6) | 61.0 (+1.0)      |
>
> Zero-shot degradation. Thank you for raising this concern. As mentioned in our paper, we believe that the slight degradation in Zero-shot classification in our SGVL models is the result of fine-tuning with negative captions and our graph prediction tasks that deviate from the original VLM training objective. This phenomenon is also visible in recent works [2, 3]. Classification performance is improved if our models are tested with linear probing. For CLIP-SGVL, when using 5-shot, 10-shot linear probing the difference on classification tasks with respect to CLIP (in the same setting) changes to -0.3%, +0.1% and with 20-shot to +0.5%. We will report the N-shot results in the final version.
>
> **How do we represent location for relationship tokens?** The location of the relationships in an image are represented by the union of bounding boxes of objects comprising them.
>
> **“I don't think unsupervised training is a natural extension of this work. The main insight brought by this work is the necessity of a small amount of densely annotated data. Future directions might include generalizing this approach beyond the VL.”** Thank you for your suggestion! We absolutely agree that this is a main insight of our work. Indeed, future directions might include generalizing this approach beyond the VL domain or applying it to other types of dense structured data (such as, segmentation masks, sketch, depth maps, normal maps etc.). We would emphasize this point further in the final version.
>
> We hope the above points have clarified and addressed your concerns. We would be happy to provide any further clarifications as requested.
>
> References:
>
> [1] "Visual Spatial Reasoning", TACL'23.
>
> [2] "Teaching structured vision & language concepts to vision & language models. CVPR 2023.
>
> [3] “When and Why Vision-Language Models Behave like Bags-Of-Words, and What to Do About It?”. ICLR 2023.

---

### Official Review · Reviewer_dPsS · 2023-08-06

**Typos Grammar Style And Presentation Improvements:** NA
**Soundness:** 4

**Excitement:**

4: Strong: This paper deepens the understanding of some phenomenon or lowers the barriers to an existing research direction.

**Missing References:**

NA

**Paper Topic And Main Contributions:**

The paper is about enhancing structural understanding of images by Vision Language Models(VLMs) using additional signals contained in scene graphs associated with the image. Since scene graph annotation is a costly process, the authors specifically look at whether small scene graph datasets can provide sufficient information to the VLMs during training/finetuning. The authors present a novel way to incorporate scene graph information into visual and textual representations by using adaptive scene graph tokens, positive and negative captions, modifications to the transformer architecture and introduce related losses. The authors try their methodology on various popular VLMs/Datasets and present encouraging results.

**Questions For The Authors:**

NA

**Reasons To Accept:**

This paper is well written and the diagrams clearly capture what the authors intended to show in their architectural improvements. Utilizing small scene graph datasets to better enable VLMs to create more granular multimodal representations can benefit the research community immensely. The authors provide comprehensive ablation studies and test their methodology using popular VLMs/Datasets. The loss functions and the architectural changes introduced are uncomplicated and easy to follow/implement.

**Reasons To Reject:**

It would have been better if more scene graph datasets were considered other than VG and 1% of LAION.

**Reproducibility:**

4: Could mostly reproduce the results, but there may be some variation because of sample variance or minor variations in their interpretation of the protocol or method.

**Reviewer Confidence:**

4: Quite sure. I tried to check the important points carefully. It's unlikely, though conceivable, that I missed something that should affect my ratings.

---

> ### Author Rebuttal · Authors · 2023-08-27
>
> ### Shared Comments
>
> We thank the reviewers for their insightful comments. We are encouraged that they found the proposed idea of leveraging small scene graph datasets **could benefit the research community immensely** `(dPsS, VrYK)` and **could successfully compensate for the lack of compositional understanding** `(VrYK)`. Furthermore, they found the proposed framework **reasonable and effective** `(639u)` and the architectural changes and losses **easy to implement** `(dPsS)`. Finally, they observed **comprehensive ablations** `(639u, dPsS)`, demonstrating **significant improvements** over the proposed tasks `(639u)`. We are pleased that the reviewers found the paper **well written** `(639u, dPsS)`, and the **diagrams capture** the novelty aspect.
>
> A brief summary of the main aspects of our rebuttal response:
> 1) We performed additional experimental comparisons to other methods, including SOTA VLMs `(639u)`.
>
> 2) We discussed the comparison between our work and other related works and explained in more detail our novelty compared to other works `(639u)`.
>
> 3) We performed a comprehensive analysis of the degradation in performance in certain partitions of the evaluated benchmark and the zero-shot degradation `(VrYK)`.
>
> 4) We performed an experiment replacing 1% of LAION with CC3M for training `(dPsS)`.
>
> We next address all concerns and look forward to an open and constructive discussion with the reviewers.
>
>
> ### Response to dPsS
>
>
> We thank the reviewer for the insightful comments. In the following, we provide a response to the questions raised in the review:
>
> **Consideration of more datasets for fine tuning**. Thank you for this comment! We present results below of our method for BLIP/BLIP2 when using the training set of the COCO dataset, instead of the LAION dataset as reported in the paper. We note that this COCO training data is roughly of the same size as the ~1% of the LAION dataset used in our submission. It can be seen that the results are similar to those reported in the submission.
>
> |                    | Winoground Text | Winoground Image | Winoground Group | VL-Checklist Attribute | VL-Checklist Object | VL-Checklist Relation |
> |--------------------|:---------------:|:----------------:|:----------------:|:----------------------:|:-------------------:|:---------------------:|
> |        BLIP        |       39.0      |       19.2       |       15.0       |          75.2          |         82.2        |          81.5         |
> |        BLIP2       |       42.0      |       23.8       |       19.0       |          77.8          |         84.9        |          84.9         |
> |  BLIP-SGVL (LAION) |       42.8      |       27.3       |       21.5       |          81.8          |         85.2        |          81.9         |
> | BLIP2-SGVL (LAION) |       42.8      |       28.5       |       23.3       |          81.2          |         88.4        |          88.8         |
> |  BLIP-SGVL (COCO)  |       43.8      |       26.0       |       21.3       |          81.9          |         85.5        |          82.0         |
> |  BLIP2-SGVL (COCO) |       46.0      |       26.5       |       22.8       |          81.2          |         88.8        |          88.8         |
>
> Regarding other scene-graph datasets, we can experiment with the Open Images dataset, and report results in the final version (we cannot complete this within the rebuttal timeline).

---

### Official Review · Reviewer_639u · 2023-08-10

**Soundness:** 4

**Excitement:**

3: Ambivalent: It has merits (e.g., it reports state-of-the-art results, the idea is nice), but there are key weaknesses (e.g., it describes incremental work), and it can significantly benefit from another round of revision. However, I won't object to accepting it if my co-reviewers champion it.

**Justification For Ethical Concerns:**

No ethical issues.

**Missing References:**

Yes, as I mentioned above, some key references on leveraging scene graph and using it to train VLM are missing. Not only the related work section but also the experimental comparison need to enrich them. For example:

[1] Yu, Fei, et al. "Ernie-vil: Knowledge enhanced vision-language representations through scene graphs." Proceedings of the AAAI Conference on Artificial Intelligence. Vol. 35. No. 4. 2021.

[2] Doveh, Sivan, et al. "Teaching structured vision & language concepts to vision & language models." Proceedings of the IEEE/CVF Conference on Computer Vision and Pattern Recognition. 2023.

[3] Ma, Zixian, et al. "CREPE: Can Vision-Language Foundation Models Reason Compositionally?." Proceedings of the IEEE/CVF Conference on Computer Vision and Pattern Recognition. 2023.

**Paper Topic And Main Contributions:**

This paper proposes an interesting vision-language pre-training method, which leverages the scene graph pairs to jointly train the text and image encoder.
Technically, the impact of scene graph is reflected from the adaptive tokens and the prompts.
In the output stage, it can additional yield the scene graph output.
The framework is pre-trained on LAION, and benchmarked on four scene understanding datasets and one zero-shot classification dataset.
Compared with conventional VLM, such as CLIP and BLIP, it achieves a better performance.

**Questions For The Authors:**

I would appreciate it if the authors can address the questions below, in the rebuttal stage:

**Q1:** Using scene graph, or more boardly structured information, is not a very novel idea now. The difference of the proposed method against some prior works need to be clarified.

**Q2:** More state-of-the-art comparison, including these latest methods and more advanced VLM.

**Reasons To Accept:**

- Techniqually, the proposed framework is reasonable and effective.

- Compared with conventional VLM such as CLIP, BLIP, it shows a significant improvement on the proposed task.

- This paper is well-written and easy-to-follow.

- The ablation studies are very extensive.

**Reasons To Reject:**

- The idea to leverage scene graph for vision-language model is not very enough now. In the past few years, there are already some references. Unfortunately, the authors do not discuss their diffference in the reference, which may in turn negatively impact the novelties of this work. For example:

[1] Yu, Fei, et al. "Ernie-vil: Knowledge enhanced vision-language representations through scene graphs." Proceedings of the AAAI Conference on Artificial Intelligence. Vol. 35. No. 4. 2021.

[2] Doveh, Sivan, et al. "Teaching structured vision & language concepts to vision & language models." Proceedings of the IEEE/CVF Conference on Computer Vision and Pattern Recognition. 2023.

[3] Ma, Zixian, et al. "CREPE: Can Vision-Language Foundation Models Reason Compositionally?." Proceedings of the IEEE/CVF Conference on Computer Vision and Pattern Recognition. 2023.

- Lack state-of-the-art comparison in the experimental section. Not only the above reference, but also some state-of-the-art VLM on the scene understanding task.

- Another minor issue is the presentation of this paper: I noticed the visual prediction are all put into the supplementary materials. It would be better to put several in the main text, as this is one of the key results for this paper.

**Reproducibility:**

4: Could mostly reproduce the results, but there may be some variation because of sample variance or minor variations in their interpretation of the protocol or method.

**Reviewer Confidence:**

4: Quite sure. I tried to check the important points carefully. It's unlikely, though conceivable, that I missed something that should affect my ratings.

**Typos Grammar Style And Presentation Improvements:**

Just one presentation issue:

The visual prediction are all put into the supplementary materials. It would be better to put several in the main text, as this is one of the key results for this paper.

---

> ### Author Rebuttal · Authors · 2023-08-27
>
> ### Shared Comments
>
> We thank the reviewers for their insightful comments. We are encouraged that they found the proposed idea of leveraging small scene graph datasets **could benefit the research community immensely** `(dPsS, VrYK)` and **could successfully compensate for the lack of compositional understanding** `(VrYK)`. Furthermore, they found the proposed framework **reasonable and effective** `(639u)` and the architectural changes and losses **easy to implement** `(dPsS)`. Finally, they observed **comprehensive ablations** `(639u, dPsS)`, demonstrating **significant improvements** over the proposed tasks `(639u)`. We are pleased that the reviewers found the paper **well written** `(639u, dPsS)`, and the **diagrams capture** the novelty aspect.
>
> A brief summary of the main aspects of our rebuttal response:
> 1) We performed additional experimental comparisons to other methods, including SOTA VLMs `(639u)`.
>
> 2) We discussed the comparison between our work and other related works and explained in more detail our novelty compared to other works `(639u)`.
>
> 3) We performed a comprehensive analysis of the degradation in performance in certain partitions of the evaluated benchmark and the zero-shot degradation `(VrYK)`.
>
> 4) We performed an experiment replacing 1% of LAION with CC3M for training `(dPsS)`.
>
> We next address all concerns and look forward to an open and constructive discussion with the reviewers.
>
>
> ### Response to R639u
>
>
> We thank the reviewer for the insightful comments. In the following, we provide a response to the questions raised in the review:
>
> **Novelty and Related work**. Thank you for suggesting these prior works. We will incorporate the discussion and results below into the final version.
>
> The main insight of our work, as highlighted by `dPsS` and `VrYK`, is that the standard VLM architecture can be modified such that a small amount of richly annotated scene graph data can successfully compensate for the lack of compositional understanding in popular pretrained VLMs. We show that roughly 100K scene graph annotations from Visual Genome, used with a specialized model architecture and a new finetuning scheme, can improve the granular understanding of pretrained VLMs. This differs from the previous works mentioned by the reviewer.
>
>
> More specifically, previous work [1] used scene graphs (parsed from texts) for masked multi-modal learning focused on objects, attributes, and relations. The approach used cross-modal transformer encoders without changing the encoder architecture itself. We, on the other hand, consider a novel architecture design for the vision component of dual-encoders that is explicitly targeted at predicting complete scene graphs from images alone. Our new architecture can be incorporated into a variety of vision-language models, and we show that it results in empirical improvement when used for fine-tuning existing models. In addition, to the best of our knowledge, we are the first to show that a small amount of scene-graph data can be used to improve the granularity of vision-language models via fine-tuning. Finally, [1] uses scene graphs that are parsed from textual captions alone, while our method emphasizes the use of dense and rich scene graph annotations from dedicated scene graph datasets such as Visual Genome.
>
> Moreover, we also differ significantly from the works described in [2] and [3]. [2] proposed several text augmentations (*no scene graphs are used*), specifically intended to enforce structured understanding on *only the textual level*. In contrast, our work exploits scene graphs on both the *visual and language sides*. In particular, our approach utilizes scene graphs to enforce structured understanding *using the visual side* (e.g., predicting scene graphs), which is not the case in [2]. Lastly, [3] proposed a new benchmark by constructing a dataset from scene graphs based on Visual Genome (which we used for training in our work). This differs from our approach, as this *work only proposed a benchmark, and not a new training method*, while our approach proposes a method for fine-tuning pretrained VLMs using scene graphs.
>
>
> **Comparison to state-of-the-art**. Following your comment, we present below additional comparisons to state-of-the-art VLMs. Specifically, SVLC [2], InstructBLIP, LLaVa, Mini-GPT4, FLAVA, and ViLT (the latter two were reported in Table 7 of the supplementary) on Winoground and VL-Checklist. We used results from papers where available and ran the models ourselves otherwise. We note that we did not report results for ERNIE-ViL, as it is not currently state-of-the-art for VL. It can be seen that our approach (SGVL) outperforms the other baselines except in one category.
>
>
> |              | Winoground Text | Winoground Image | Winoground Group | VL-Checklist Attribute | VL-Checklist Object | VL-Checklist Relation |
> |--------------|:---------------:|:----------------:|:----------------:|:----------------------:|:-------------------:|:---------------------:|
> |   SVLC [2]   |       28.0      |       8.75       |       5.75       |          69.3          |         82.3        |          82.1         |
> |     LLaVa    |       24.8      |       25.0       |       13.0       |          65.5          |         83.1        |          83.0         |
> |   Mini-GPT4  |       23.3      |       18.0       |        9.5       |          71.3          |         84.2        |          84.1         |
> | InstructBLIP |       38.0      |       25.0       |       19.8       |          76.7          |         84.5        |        **91.0**       |
> |     FLAVA    |       32.3      |       20.0       |       14.5       |          54.6          |         70.6        |          46.6         |
> |     ViLT     |       34.8      |       14.0       |        9.3       |          73.3          |         85.0        |          62.0         |
> |   CLIP-SGVL (Ours)  |       32.0      |       14.0       |        9.8       |          72.0          |         82.6        |          82.0         |
> |   BLIP-SGVL (Ours)  |       42.8      |       27.3       |       21.5       |        **81.8**        |         85.2        |          81.9         |
> |  BLIP2-SGVL (Ours)  |     **42.8**    |     **28.5**     |     **23.3**     |          81.2          |       **88.4**      |          88.8         |
>
> **Presentation of the visual prediction**. Thank you for this suggestion! Following your comment, we will include the visual prediction in the main paper from the supplementary materials, as the camera ready includes a 9-page limit, one more page than the current draft.
>
> **Reproducibility**. It is important to note that we have included our code base in the supplementary, and thus the results should be reproducible. We will also release the weights and models publicly upon acceptance.
>
> We hope the above points have clarified and addressed your concerns. We would be happy to provide any further clarifications as requested.
>
>
>
>
> References:
>
> [1] Yu, Fei, et al. "Ernie-vil: Knowledge enhanced vision-language representations through scene graphs." Proceedings of the AAAI Conference on Artificial Intelligence. Vol. 35. No. 4. 2021.
>
> [2] Doveh, Sivan, et al. "Teaching structured vision & language concepts to vision & language models." Proceedings of the IEEE/CVF Conference on Computer Vision and Pattern Recognition. 2023.
>
> [3] Ma, Zixian, et al. "CREPE: Can Vision-Language Foundation Models Reason Compositionally?." Proceedings of the IEEE/CVF Conference on Computer Vision and Pattern Recognition. 2023.

---

### Meta-Review · Area_Chair_6Qzj · 2023-09-09

**Recommendation:** 5

**Metareview:**

The paper aims to use scene graph to improve the VL models. The scene graph supervises the vision branch via two tasks: 1. direct supervise with token prediction. 2. serves to construct hard negatives in the language branch. The overall paper quality is good.

---

### Decision · Program_Chairs · 2023-10-07

**Decision:**

Accept-Main

**Comment:**

The paper aims to use scene graph to improve the VL models. The scene graph supervises the vision branch via two tasks: 1. direct supervise with token prediction. 2. serves to construct hard negatives in the language branch. The overall paper quality is good.